# Data Value in the Age of Scaling: Understanding LLM Scaling Dynamics Under Real–Synthetic Data Mixtures

## Abstract

The rapid progress of large language models (LLMs) is fueled by the growing reliance on datasets that blend real and synthetic data. While synthetic data offers scalability and cost-efficiency, it often introduces systematic distributional discrepancies, particularly underrepresenting long-tail knowledge due to truncation effects from data generation mechanisms like top-$p$ sampling, temperature scaling, and finite sampling. These discrepancies pose fundamental challenges in characterizing and evaluating the utility of mixed real-synthetic datasets. In this paper, we identify a three-phase scaling behavior characterized by two breakpoints that reflect transitions in model behavior across learning head and tail knowledge. We further derive an LLM generalization bound designed for real and synthetic mixtures, revealing several key factors that govern their generalization performance. Building on our theoretical findings, we propose an effective yet efficient data valuation method that scales to large-scale datasets. Comprehensive experiments across four tasks, including image classification, sentiment classification, instruction following, and complex reasoning, demonstrate that our method surpasses state-of-the-art baselines in data valuation with significantly low computational cost.

## 1 Introduction

Large language models (LLMs) have achieved remarkable advances, driving unprecedented transformations across various tasks, including language understanding (Wong et al., 2024), generation (Wong et al., 2024), instruction following (Lou et al., 2024), and reasoning (Plaat et al., 2024). Despite these achievements, their performance is largely driven by the scale and quality of training datasets (Brown et al., 2020; Hoffmann et al., 2022). To mitigate the scarcity and high cost of high-quality real data, many modern training pipelines incorporate synthetically generated data, which can be scaled efficiently through data augmentation or controlled generation (Thakur et al., 2023; Zhang et al., 2024b). While synthetic data plays a critical role in scaling data at reduced cost, it often introduces systematic distributional discrepancies, resulting in unintended negative impacts on model performance (Chen et al., 2024b). In particular, synthetic datasets inherently bias training towards frequently occurring knowledge while neglecting rare but significant knowledge (Seddik et al., 2024). Consequently, such discrepancies can degrade the overall generalization capabilities of LLMs on downstream tasks, leading to model collapse and failure to capture underrepresented knowledge (Shumailov et al., 2024).

One potential explanation for this challenge lies in the inherent long-tail distribution of knowledge present in real-world data. Empirical studies have shown that real-world knowledge typically follows a long-tail distribution, where a small amount of prevalent ("head") knowledge appears frequently, while numerous rare ("tail") knowledge occur infrequently but collectively represent a significant portion of essential knowledge (Zhang et al., 2024c), as shown by the orange curve in Figure 1. For example, large language models usually perform well on general questions (e.g., normal disease diagnosis) but struggle when answering rare or highly specific questions (e.g., rare disease diagnosis) (Kandpal et al., 2023). Synthetic data generation methods often exacerbate this imbalance in the distribution of knowledge because their inherent generation biases towards common knowledge make rare knowledge even more scarce in the training data. As a result, LLMs trained on datasets of real and synthetic mixtures exhibit complex scaling behaviors, reducing learning efficiency and generalization in pre-training and fine-tuning steps. These observations motivate us to ask two fundamental research questions: **(Q1)** What are the scaling behaviors of large language models when

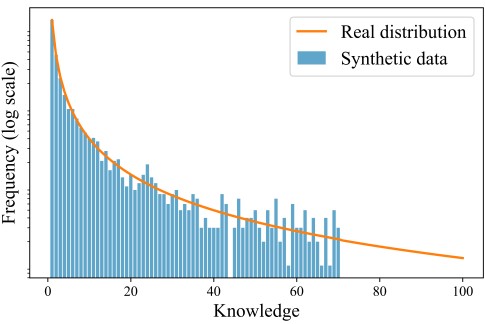

Figure 1: The real-world knowledge follows a long-tail distribution (illustrated with the greatest common divisor task (Charton, 2023)). Synthetic data is often sampled only from the head knowledge, leading to a truncated tail.

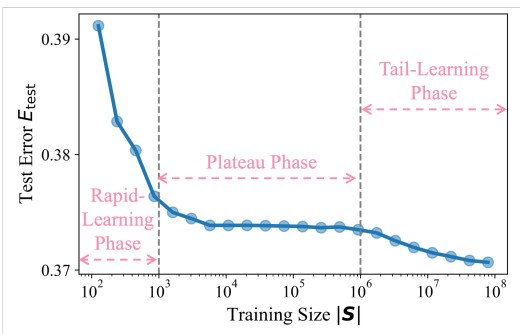

Figure 2: Fine-grained three-phase scaling behavior on real and synthetic mixtures, illustrated with the greatest common divisor task (Charton, 2023).

trained on real and synthetic mixtures, and how do these behaviors impact the acquisition of tail knowledge? and **(Q2)** How can we develop an salient data valuation framework to identify valuable subsets of data, thereby better guiding the training process under real and synthetic mixtures?

To address the first question **(Q1)** on the scaling behaviors of LLMs trained on real–synthetic data mixtures, we identify a *three-phase scaling pattern* in the training process of LLMs, as illustrated in Figure 2. In particular, an initial *Rapid-Learning Phase* dominated by frequent ("head") knowledge present abundantly in both real and synthetic data; a subsequent *Plateau Phase*, in which additional data provides diminishing returns due to the limited coverage of rare ("tail") knowledge in synthetic data; and a final *Tail-Learning Phase*, where sufficient real data containing the tail knowledge enables further performance gains. We further introduce a novel theoretical framework based on the LLM generalization bound. This framework reveals the generalization error in terms of empirical losses of real and synthetic mixtures, the distribution discrepancies between training and test distributions, the neural tangent kernel (NTK (Jacot et al., 2018)) reflecting training dynamics, and the proportion of real data in the training set.

To empirically guide the LLM training process under real and synthetic mixtures and address the second research question **(Q2)**, we propose a scalable and theoretically grounded data valuation framework. Traditional data valuation techniques, such as Leave-One-Out (LOO (Koh & Liang, 2017)) and Shapley Values (SV (Ghorbani & Zou, 2019)), require retraining the model multiple times on different subsets, which is computationally infeasible for models with millions of parameters (Koh & Liang, 2017; Jia et al., 2019). Our proposed data valuation framework is directly derived from our LLM generalization bound, enabling computationally efficient and theoretically grounded estimation of the contributions of individual data subsets without retraining, thereby potentially improving training efficiency to guide the training process under real and synthetic mixtures.

Finally, we empirically validate both our theoretical findings and the effectiveness of the proposed data valuation method through extensive experiments. Specifically, we evaluate our framework across four representative tasks, covering image classification, sentiment classification, instruction-following, and complex reasoning. Notably, we observe the predicted three-phase scaling behavior in an image classification task explicitly characterized by a known long-tail distribution. Furthermore, experimental results demonstrate that our valuation method outperforms existing baselines in effectively identifying high-value data subsets with a low computation cost. In particular, our valuation scores exhibit the highest correlation with ground-truth compared to the baseline methods, peaking at $\sim 20\times$ in the strongest case. We open-source our code at the anonymous link.

## 2 PRELIMINARY

In this section, we introduce the background that is pertinent to our work. Next, we briefly review notations, LLM scaling law, and LLM generalization.

**Notations.** Modern LLMs are increasingly trained on datasets composed of real and synthetic mixtures. Let $S = S_1 \cup S_2$ denote the training dataset, where $S_1 \sim \mathcal{D}$ consists of real data drawn from the true distribution $\mathcal{D}$, and $S_2 \sim \mathcal{D}'$ consists of synthetic data generated by model with an

associated distribution $\mathcal{D}'$. We assume that the overall training distribution can be written as:

$$\mathcal{D}_S = \pi \mathcal{D} + (1 - \pi)\mathcal{D}', \tag{1}$$

where $\pi \in [0, 1]$ is the proportion of real data in the training set. Suppose the total number of training samples is $|\boldsymbol{S}|$, then $\pi|\boldsymbol{S}|$ samples are drawn from $\mathcal{D}$ and $(1 - \pi)|\boldsymbol{S}|$ from $\mathcal{D}'$. Model performance is evaluated on a test set $\boldsymbol{T}$ of size $|\boldsymbol{T}|$, drawn from the distribution $\mathcal{D}_T$. Let $\mathcal{L}_{\boldsymbol{S}}(f)$ denote the empirical error of model $f$ on dataset $\boldsymbol{S}$, and $\mathcal{L}_{\mathcal{D}_T}(f)$ denote its generalization error on $\mathcal{D}_T$.

**LLM Scaling Law.** Scaling laws reveal how model performance improves with increasing dataset size, model parameters, and computational resources and guide large-scale training strategies (Kaplan et al., 2020; Hoffmann et al., 2022; Hernandez et al., 2021). In practical scenarios, a critical challenge arises from the reliance on synthetic data, which may lack the coverage of real-world data distribution. This reliance can lead to model collapse: as the model fits more synthetic samples, it reinforces biases from synthetic data $\mathcal{D}'$, exhibiting severe generalization degradation relative to the true distribution $\mathcal{D}$ (Shumailov et al., 2024; Dohmatob et al., 2024b;c; Jain et al., 2024). Recent efforts attempt to extend scaling laws under real and surrogate data, but typically put strong modelling assumptions. For example, a common design draws independent samples from real and synthetic distributions that both belong to the Gaussian distribution $\boldsymbol{x} \sim \mathcal{N}(\mu, \boldsymbol{\Sigma})$, with different parameters. However, these efforts often overlook the long-tail nature of real-world knowledge.

**LLM Generalization.** To theoretically understand the LLM generalization, the neural tangent kernel has emerged as a powerful analytical framework for characterizing the training dynamics of neural networks with gradient descent (Jacot et al., 2018). Consider a $L$-layer LLM with $m_l$ parameters in layer $l = 1, \ldots, L$. Following prior literature (Lee et al., 2019), we assume $m_1 = \cdots = m_{L-1} = m$ and $m_L = 1$ to simplify our analysis. Based on the formulation above, the NTK $\Theta \in \mathbb{R}^{|\boldsymbol{S}| \times |\boldsymbol{S}|}$ of a model $f(\boldsymbol{x}; \boldsymbol{\theta})$ on the dataset $\boldsymbol{S}$ is defined as

$$\boldsymbol{\Theta}(\boldsymbol{x}, \boldsymbol{x}'; \boldsymbol{\theta}) = \nabla_{\boldsymbol{\theta}} f(\boldsymbol{x}; \boldsymbol{\theta})^\top \nabla_{\boldsymbol{\theta}} f(\boldsymbol{x}'; \boldsymbol{\theta}), \tag{2}$$

where $\boldsymbol{x}$ (or $\boldsymbol{x}'$) denotes any data point in dataset $\boldsymbol{S}$. Interestingly, as $m_1, \ldots, m_{L-1} \to \infty$, the NTK $\boldsymbol{\Theta}_0$ based on the initialized model parameters $\boldsymbol{\theta}_0$ will finally converge to a deterministic form $\boldsymbol{\Theta}_\infty$ (Jacot et al., 2018; Yang & Littwin, 2021; Cao & Gu, 2019). However, existing LLM generalization bounds do not explicitly account for training on real and synthetic mixtures.

**Problem Definition.** The goal of this paper is to analyze LLMs under real and synthetic mixtures from two complementary perspectives. In particular, given the training set $\boldsymbol{S}$ contain $\pi|\boldsymbol{S}|$ samples from true distribution $\mathcal{D}$ and $(1 - \pi)|\boldsymbol{S}|$ from synthetic distribution $\mathcal{D}'$, how can we (1) theoretically reveal the scaling behavior of LLM model $f$ as detailed in Section 3? and (2) how can we develop a data valuation framework that estimates the contribution of each data subset in $\boldsymbol{S}$ to the model's performance as detailed in Section 4?

# 3 THEORETICAL ANALYSIS

In this section, we first analyze a fine-grained three-phase transition in the scaling behavior of LLMs when trained on real and synthetic mixtures. We then derive a novel LLM generalization bound for real and synthetic mixtures, which reveals four key factors that govern the generalization performance.

**Three Phase Transitions.** To understand the scaling behaviors of LLMs when trained on real and synthetic mixtures and how these behaviors impact the acquisition of tail knowledge **(Q1)**, we analyze the behavior of LLMs under a realistic training setup. While prior work has investigated scaling behaviors in the context of model collapse, these studies (Feng et al., 2024; Dohmatob et al., 2024a;b;c) often rely on strong assumptions about model and data distributions (*e.g.*, deterministic settings, simplified linear regression models, or infinite original samples). In contrast, we consider a practical scenario where the knowledge $i$ in real data exhibits a long-tail distribution $\mathcal{D}$. In natural language datasets, the word or token frequencies often exhibit long-tail distributions (Zipf's law (Zipf, 2013)), which means a few "head" tokens occur extremely frequently, while many "tail" tokens appear rarely. We therefore model the true distribution $\mathcal{D}$ over knowledge $i$ by:

$$p_i \propto i^{-\beta}, \quad i = 1, 2, \ldots, \tag{3}$$

where $\beta > 1$ characterizes the tail heaviness. Furthermore, when generating synthetic data via LLMs, the resulting data distribution $\mathcal{D}'$ typically exhibits truncation in the tail. Specifically, the techniques of synthetic data generation inherently truncate or narrow the original distribution of generated

tokens, thereby cutting off or diminishing probabilities for less frequent (tail) tokens (Dohmatob et al., 2024c). For example, top-$p$ (nucleus) sampling, where tokens beyond a cumulative probability threshold are discarded; temperature scaling, which modifies the probability distribution sharpness; or finite-sample biases, which restrict observation of low-frequency tokens. We assume the synthetic data distribution $p'$ mirrors the true distribution $p$ up to a finite cutoff $k$: $p'_i \propto i^{-\beta}$ for $i \leq k$, and $p'_i = 0$ for $i > k$. Therefore, the training dataset of total size $|\boldsymbol{S}|$ is composed of real data $p_i$ with proportion $\pi$ and synthetic data $p'_i$ with proportion $(1 - \pi)$, where data is drawn from the distribution $\mathcal{D}_S$ with probability:

$$q_i = \pi p_i + (1 - \pi)p'_i. \tag{4}$$

We further assume that if knowledge $i$ is observed in the training set, it is predicted correctly with probability $\rho(i) = ai^{-\alpha}, a > 0$; if knowledge $i$ is not observed, the probability is $\gamma(i) = bi^{-\lambda}, b > 0$. Under the setting, we establish the following lemma for the test error on $\mathcal{D}_T$ of this model with respect to the true data distribution $\mathcal{D}_S$: $\mathcal{L}_{\text{test}} = \mathbb{E}_{(\boldsymbol{x},y)\sim\mathcal{D}_T}[\ell(f_{\mathcal{D}_S}(\boldsymbol{x}), y)]$, where $f_{\mathcal{D}_S}$ is the model on $\mathcal{D}_S$ and $\ell$ is the loss function:

**Lemma 1** (Scaling Behavior with Three phases). *Consider training data where the probability of knowledge $i$ is $q_i = \pi p_i + (1 - \pi)p'_i$, where $p_i \propto i^{-\beta}$ and $p'_i$ is cut off at rank $k$ as defined above. The test error $\mathcal{L}_{test}$ exhibits distinct scaling regimes characterized by two breakpoints at sample sizes $|\boldsymbol{S}| = k^\beta$ and $|\boldsymbol{S}| = k^\beta/\pi$. We have[1]:*
***Phase 1 (Rapid-Learning):*** *$|\boldsymbol{S}| \leq c_1 k^\beta$, where $c_1$ is absolute constant,*

$$\mathcal{L}_{test} \asymp a\,|\boldsymbol{S}|^{\frac{1-\alpha-\beta}{\beta}} - b\,|\boldsymbol{S}|^{\frac{1-\lambda-\beta}{\beta}} + a\,k^{1-\alpha-\beta} - b\,k^{1-\lambda-\beta} + k^{1-\beta}. \tag{5}$$

***Phase 2 (Plateau):*** *$c_1 k^\beta < |\boldsymbol{S}| < c_2 k^\beta/\pi$, where $c_2$ is absolute constant, $\mathcal{L}_{test}$ enters a transition state as the limited presence of tail knowledge prevents the rapid learning.*
***Phase 3 (Tail-Learning):*** *$|\boldsymbol{S}| \geq c_2 k^\beta/\pi$,*

$$\mathcal{L}_{test} \asymp a(\pi|\boldsymbol{S}|)^{\frac{1-\alpha-\beta}{\beta}} - b(\pi|\boldsymbol{S}|)^{\frac{1-\lambda-\beta}{\beta}} + k^{1-\beta}. \tag{6}$$

**Remark #1:** For frequently occurring (head) knowledge indexed by 1 through $k$, the performance scaling exhibits a critical transition at sample size $|\boldsymbol{S}| = k^\beta$, corresponding to the first breakpoint in Figure 3.

**Remark #2:** For infrequently occurring (tail) knowledge beyond rank $k$, the performance scaling exhibits a critical transition at sample size $|\boldsymbol{S}| = k^\beta/\pi$, corresponding to the second breakpoint Figure 3.

**Remark #3:** This lemma highlights three phases of performance improvement as training size $|\boldsymbol{S}|$ grows. As shown in Figure 3, initially in the rapid-learning phase, rapid performance gains occur predominantly due to extensive coverage and repeated sampling of

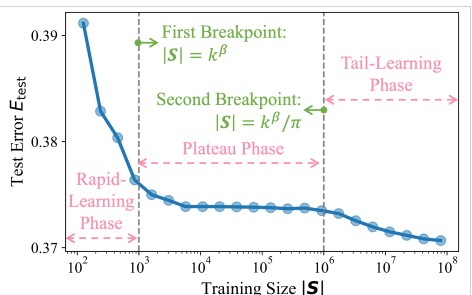

Figure 3: Three-phase scaling behavior with two breakpoints on real–synthetic mixtures, for the same task as Figure 2.

head knowledge, supported by both real and synthetic data. As head knowledge becomes saturated, a plateau phase follows, characterized by minimal improvements. This stagnation arises because the model gains limited additional information from redundant head knowledge, and the data distribution has not yet yielded sufficient tail-class observations. Leveraging targeted data valuation strategies (as introduced in Section 4), one can efficiently identify and prioritize underrepresented knowledge, potentially improving training efficiency. Finally, in the tail-learning phase, the model's performance significantly improves again as it learns from substantial accumulated real samples of tail knowledge.

**LLM Generalization Bound.** To provide a general theoretical understanding of LLMs trained on real–synthetic data mixtures, we derive a novel generalization bound with relaxed assumptions. Existing generalization bounds typically assume that all training data are drawn i.i.d. from a single distribution (Lotfi et al., 2023). However, this assumption is overly simplistic for practical scenarios, as real-world datasets often supplement limited real datasets with synthetic data generated from large

---

[1]The notation $g(n) \asymp h(n)$ means that $c_1 h(n) \leq g(n) \leq c_2 h(n)$ for sufficiently large $n$ and absolute constants $c_1, c_2 > 0$.

models. Our LLM generalization bound reflects a realistic and growing training regime in LLMs. It explicitly quantifies how empirical losses on training data of real-synthetic mixtures, the distributional discrepancies, the NTK, and data composition collectively influence the expected test loss.

To characterize the distribution discrepancy under the setting of real and synthetic mixtures, we introduce the $\mathcal{H}$-discrepancy $d_{\mathcal{H}}$ (Definition 1) in Appendix A. To analyze the training dynamics of LLMs, we employ the NTK. Following the assumptions in Shu et al. (2022), we assume that the existence of a function class $\mathcal{H}$ such that for any $\boldsymbol{x}$, the deviation between the model $f(\boldsymbol{x}; \boldsymbol{\theta}) \in [0, 1]$ and the optimal hypothesis $f^*(\boldsymbol{x}; \boldsymbol{\theta}) = \arg\min_f (\mathcal{L}_{\mathcal{D}_T}(f) + \mathcal{L}_{\mathcal{D}_S}(f))$ is bounded by some $h \in \mathcal{H}$ with $h(x) \leq 1$. Our generalization bound is then derived based on both the NTK at initialization $\boldsymbol{\Theta}_0$ and at convergence $\boldsymbol{\Theta}_\infty$, and the distribution discrepancy between $\boldsymbol{S}_1$, $\boldsymbol{S}_2$, and $\boldsymbol{T}$:

---

**Theorem 1** (LLM Generalization Bound under Real and Synthetic Mixtures). *Let $\lambda_{\min}(\cdot)$ and $\lambda_{\max}(\cdot)$ denote the minimum and maximum eigenvalue of a matrix. Assume $\lambda_{\min}(\boldsymbol{\Theta}_0) > 0$ and $\|\nabla_{\boldsymbol{\theta}} f(\boldsymbol{x}; \boldsymbol{\theta}_0)\|_2 \leq B$ for any $(\boldsymbol{x}, y) \in \boldsymbol{S}$ with $\|\boldsymbol{x}\|_2$, $y \in [0, 1]$. There exist $M \in \mathbb{N}$ such that for every $m > M$, when applying gradient descent with learning rate $\eta < \min\left\{2m^{-1}\left(\lambda_{\min}(\boldsymbol{\Theta}_\infty) + \lambda_{\max}(\boldsymbol{\Theta}_\infty)\right)^{-1}, |\boldsymbol{S}|/\lambda_{\max}(\boldsymbol{\Theta}_0)\right\}$, with probability at least $1 - 2\delta$,*

$$\mathcal{L}_{D_T}(f) \leq \pi \mathcal{L}_{\boldsymbol{S}_1}(f) + (1 - \pi)\mathcal{L}_{\boldsymbol{S}_2}(f) + \pi d_{\mathcal{H}}(\boldsymbol{T}, \boldsymbol{S}_1) + (1 - \pi)d_{\mathcal{H}}(\boldsymbol{T}, \boldsymbol{S}_2)$$

$$+ 2B\sqrt{\frac{\hat{\boldsymbol{y}}^T \boldsymbol{\Theta}_0^{-1}\hat{\boldsymbol{y}}}{|\boldsymbol{S}|}} + \sqrt{\frac{2\max(\pi, 1 - \pi)\log(8/\delta)}{|\boldsymbol{S}|}} + \varepsilon, \tag{7}$$

*where each element in $\hat{\boldsymbol{y}}$ is defined as $\hat{y} \triangleq y - f(\boldsymbol{x}; \boldsymbol{\theta}_0)$ and $\varepsilon \triangleq 2c/\sqrt{m} + 3\sqrt{\log(4/\delta)/2|\boldsymbol{S}|} + \sqrt{\log(4/\delta)/2|\boldsymbol{T}|} + \mathcal{L}_{\mathcal{D}_T}(f^*) + \mathcal{L}_{\mathcal{D}_S}(f^*)$, and $c > 0$ is a constant.*

---

**Remark:** Theorem 1 shows that the generalization error on the test distribution is bounded in terms of four key factors: (1) the empirical loss on training real samples and training synthetic samples; (2) the distribution discrepancy between test data and train data of real and synthetic mixtures; (3) the NTK-related value at initialization; and (4) the composition of the training dataset, specifically the proportion $\pi$ of real data and the total number of samples $|\boldsymbol{S}|$.

## 4 METHOD

In this section, we introduce a data valuation framework designed for training settings involving real–synthetic data mixtures to solve **Q2**. Existing data valuation methods (Lin et al., 2024; Fleckenstein et al., 2023; Wang & Jia, 2023; Kwon & Zou, 2022; Xu et al., 2021) typically require multiple retrainings or assume that all training data is drawn from a single distribution. These methods are not scalable to large models and, more importantly, do not explicitly consider the real-world data composed of real and synthetic mixtures. Our method is derived directly from the generalization bound in Section 3, and is designed to estimate the contribution of data subsets (data contributors) under real-synthetic mixtures, while remaining retraining-free and thus scalable to LLMs.

Specifically, we realize the discrepancy $d_{\mathcal{H}}$ using multiple-kernel maximum mean discrepancy (MK-MMD (Gretton et al., 2012c)) in reproducing kernel Hilbert spaces (Long et al., 2015; Sejdinovic et al., 2012), which captures a wide class of hypotheses while retaining computational efficiency. Moreover, the use of multiple kernels enables adaptive integration of features at different scales, which is well-suited for LLM training scenarios where real and synthetic data may differ significantly in linguistic style, topical coverage, or vocabulary distribution (Gretton et al., 2012b). In data valuation, we compare the relative performances of data contributors; the constant $\varepsilon$ in Theorem 1 is independent of the ranking of data contributors. We therefore ignore $\varepsilon$ while reducing computational cost. Given a training dataset $\boldsymbol{S} = \boldsymbol{S}_1 \cup \boldsymbol{S}_2$, where $\boldsymbol{S}_1 \sim \mathcal{D}$ (real data) and $\boldsymbol{S}_2 \sim \mathcal{D}'$ (synthetic data), and a test distribution $\boldsymbol{T} \sim \mathcal{D}_T$, we define the data valuation score as (see Algorithm 1 in Appendix B):

$$v(\boldsymbol{S}) = w_1\left[\pi \mathcal{L}_{\boldsymbol{S}_1}(f) + (1 - \pi)\mathcal{L}_{\boldsymbol{S}_2}(f)\right] + w_2\left[\pi \mathrm{Dist}(\boldsymbol{T}, \boldsymbol{S}_1) + (1 - \pi)\mathrm{Dist}(\boldsymbol{T}, \boldsymbol{S}_2)\right]$$

$$+ w_3\sqrt{\frac{\hat{\boldsymbol{y}}^\top \boldsymbol{\Theta}_0^{-1}\hat{\boldsymbol{y}}}{|\boldsymbol{S}|}} + w_4\sqrt{\frac{\max(\pi, 1 - \pi)}{|\boldsymbol{S}|}}, \tag{8}$$

where $\mathcal{L}_{\boldsymbol{S}_i}(f)$ denotes the empirical loss on real ($i = 1$) or synthetic ($i = 2$) data, Dist is the MK-MMD metric (Gretton et al., 2012c), $f$ and $\boldsymbol{\Theta}_0$ are the model and empirical NTK at initialization.

$\hat{\boldsymbol{y}}$ is evaluated on dataset $\boldsymbol{S}$ following its definition in Theorem 1. $\pi$ is the proportion of real data in the training set, $w_1, w_2, w_3, w_4$ balance the contribution of the four terms.

The valuation function $v(\boldsymbol{S})$ in Eq.(8) directly reflects the components in our theoretical generalization bound. Each component of the empirical losses, distribution discrepancies, and the NTK, corresponds to a measurable quantity that influences generalization performance. This translation from theory to scoring function is particularly suited for LLMs, where large-scale training makes retraining-based valuation infeasible. The valuation function also provides a practical handle on the three-phase scaling behavior in Section 3. In the first phase, $v(\boldsymbol{S})$ highlights subsets from head classes that rapidly reduce the empirical losses. During the plateau phase, where head-class performance saturates, the NTK-based generalization term becomes critical, distinguishing data that meaningfully alters the function class from data that is redundant or uninformative. In the final phase, as tail classes begin to appear in real data, the function prioritizes examples that drive continued error reduction. Notably, our scoring function $v(\boldsymbol{S})$ is designed to be directly applicable in LLM-scale settings, but it also supports integration with marginal-contribution-based valuation methods, see Appendix C for details.

## 5 EXPERIMENTS

In this section, we evaluate the effectiveness of our data valuation method under datasets of real and synthetic mixtures. We conduct experiments across four representative tasks: image classification, sentiment classification, instruction following, and complex reasoning. As detailed in Section 5.2, we first verify that the three-phase generalization behavior predicted by our theoretical analysis emerges in practice under a controlled long-tail setting. Section 5.3 compares our method against five recent data valuation baselines across all tasks and various backbones. Our method achieves higher correlation with ground-truth while maintaining significantly low computational cost. Finally, Section 5.4 demonstrates that the relative values computed using our scoring function remain stable under subsampling, supporting the scalability of our framework for large-scale LLM tasks. Beyond the main results, we include an extended analysis of contributors' ranking visualization across data valuation methods in Appendix E.

### 5.1 EXPERIMENTAL SETUP

**Tasks and Datasets.** We consider the following four tasks: (1) *Image Classification* is the task of assigning a label to a given image. We use the CIFAR-100 dataset (Krizhevsky et al., 2009) as the real data, and generate synthetic data by applying corruption transformations from the CIFAR-100-C benchmark (Hendrycks & Dietterich, 2019). (2) *Sentiment Classification* is the task of determining the sentiment polarity (positive or negative) of a given text, such as a movie review. We use the IMDb (Maas et al., 2011) as the real dataset and the FinGPT Sentiment Train dataset (Yang et al., 2023) as synthetic data. (3) *Instruction Following* involves generating an appropriate response or action based on a natural language instruction, testing a model's ability to comprehend and execute commands or answer questions accurately. We use the Natural-Instructions dataset (Mishra et al., 2021) as the real dataset and the Magpie-Pro-1M dataset (Xu et al., 2024) as the synthetic dataset. (4) *Complex Reasoning*, particularly in mathematical problem-solving, requires generating multi-step reasoning processes to arrive at a solution, often using a technique called chain-of-thought (CoT) reasoning, where the model breaks down a problem into intermediate steps before providing the final answer. We use the human-annotated portions of the NuminaMath-CoT training set (Li et al., 2024) as real data and the synthetically generated portions as synthetic data.

**Baselines.** We compare against five representative baselines designed for efficient data valuation: *DAVINZ* (Wu et al., 2022), *Deviation* (Lin et al., 2024), *LOGRA* (Choe et al., 2024), *TracIn* (Pruthi et al., 2020), and *TRAK* (Park et al., 2023). These baselines are selected based on two criteria: (1) they do not require repeated model retraining, making them scalable to LLMs; and (2) they operate with access to checkpoints, gradients, and training/test data.

**Implementation Details.** For all tasks, each method receives the same inputs: training data (real and synthetic), test data, model checkpoints, and access to model gradients. Due to the large-scale nature of LLM, we compute gradients for only 1% of the training data when evaluating gradient-based baselines to reduce computational overhead and improve efficiency. We use ResNet-18 for image classification task. For sentiment classification, instruction following, and complex reasoning tasks, we consider four backbones, including Qwen2.5-0.5B, Qwen3-0.6B, Qwen3-1.7B, and Llama-3.2-1B-Instruct. We use the Pearson, Spearman, and Kendall correlations between the data valuation scores and the ground truth as evaluation metrics. Following prior work (Wu et al., 2022), we use ground truth to refer to the test performance of models trained to convergence on different subsets of

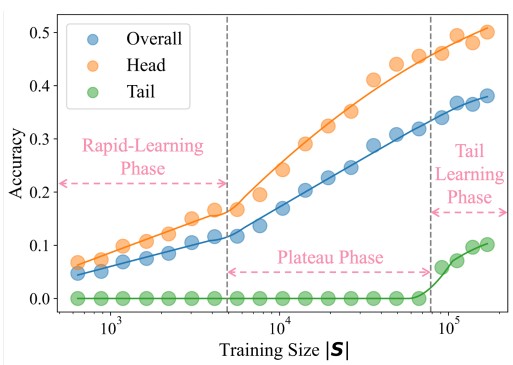
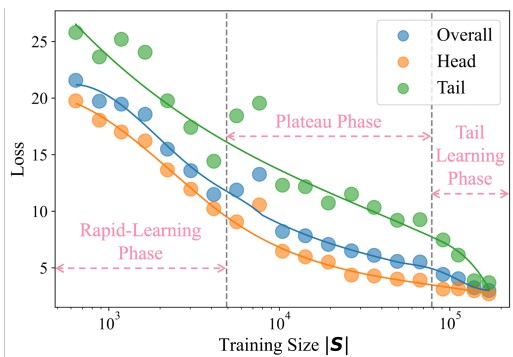

Figure 4: Model accuracy as the increase of training size $|\boldsymbol{S}|$ on CIFAR-100, under a long-tail class distribution. Dashed grey lines mark predicted transition breakpoints at $|\boldsymbol{S}| = k^\beta$ (left) and $|\boldsymbol{S}| = k^\beta/\pi$ (right).

Figure 5: Test loss as the increase of training size $|\boldsymbol{S}|$ on CIFAR-100, under a long-tail class distribution. Dashed grey lines mark predicted transition breakpoints at $|\boldsymbol{S}| = k^\beta$ (left) and $|\boldsymbol{S}| = k^\beta/\pi$ (right).

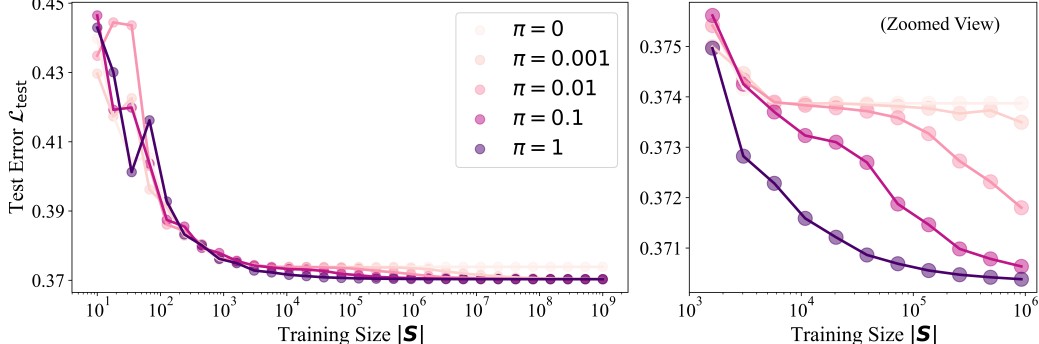

Figure 6: Simulation of three-phase scaling behavior under real-synthetic mixtures. Each curve represents a different mixture ratios of real and synthetic data $\pi$. The right panel shows a zoomed view of the range $|\boldsymbol{S}| \in [10^3, 10^6]$.

data. Specifically, the ground truth represents test accuracy for image classification and sentiment classification tasks, IFEval score for instruction following, and correctness for complex reasoning achieved by fully trained models, where each model is trained using data from different contributors. Further details about the experimental setups are provided in Appendix D.

## 5.2 VALIDATING THEORETICAL ANALYSIS

To empirically validate our theoretical insights on the three-phase scaling behavior in Section 3, we conduct experiments using CIFAR-100 as the real data and its corrupted variant (CIFAR-100-C) as the synthetic data. The proportion of real data is set to $\pi = 0.0625$, and we vary the total training sample size from $10^2$ to $10^6$. We treat each of the 100 classes as one knowledge. To simulate a long-tail distribution, we manually construct a class frequency with $p_i \propto i^{-2}$ and apply a tail cutoff at $k = 70$. The model backbone is ResNet-18. We evaluate the test performance on a balanced test set with $10,000$ samples, measuring both accuracy and loss separately for the overall classes, head classes ($i \leq 70$), and tail classes ($i > 70$). Figures 4 and 5 plot the model's accuracy and test loss, respectively, as the increase of training sample size $|\boldsymbol{S}|$. The results exhibit a three-phase behavior consistent with our theoretical predictions: Phase 1 (rapid-learning), we observe a sharp decrease in head-class loss, indicating that the model quickly learns head knowledge from both real and synthetic data. Phase 2 (plateau), the overall loss reduction slows down, reflecting diminishing returns from saturated head information. Phase 3 (tail-learning), tail-class accuracy improves and loss drops rapidly, as the model learns tail knowledge from the increased number of real data.

In addition, we further validate our theoretical results with respect to diverse mixture ratios $\pi$ ranging from 0 to 1. The knowledge follows a long-tail distribution with $\beta = 1.5$ and tail cutoff $k = 100$. The model predicts a knowledge $i$ correctly with probability $\rho(i) = i^{-0.5}$ if observed and $\gamma(i) = i^{-1}$ if unobserved. Figure 6 demonstrates that the three-phase scaling behavior holds consistently across different mixture ratios of real and synthetic data.

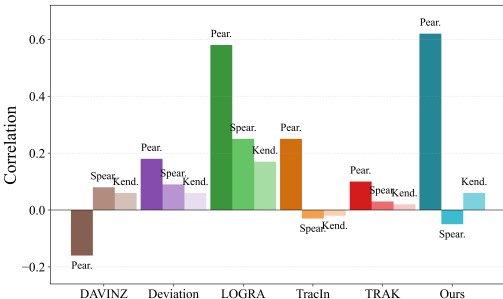 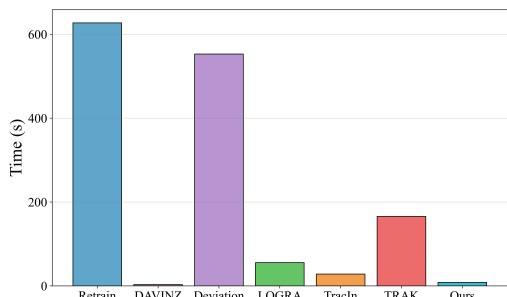

Figure 7: Comparison of data valuation methods on the image classification task. We report Pearson, Spearman, and Kendall correlations, with higher values indicating better performance.

Figure 8: Runtime (in seconds) of data valuation methods on the image classification task. The reported values represent the average time for all data contributors.

Table 1: Comparison of data valuation methods across three tasks: sentiment classification, instruction following, and complex reasoning. For each task, we report the Pearson, Spearman, and Kendall correlations, where higher is better ($\uparrow$). The best results are shown in bold.

| Backbone | Method | Sentiment | | | Instruction | | | Reasoning | | |
|---|---|---|---|---|---|---|---|---|---|---|
| | | Pear. | Spear. | Kend. | Pear. | Spear. | Kend. | Pear. | Spear. | Kend. |
| | DAVINZ | -0.46 | -0.42 | -0.33 | -0.16 | -0.40 | -0.33 | -0.00 | -0.02 | -0.01 |
| | Deviation | 0.63 | 0.76 | 0.56 | 0.05 | -0.20 | 0.00 | -0.03 | 0.00 | -0.00 |
| Qwen2.5-0.5B | LOGRA | -0.64 | -0.79 | -0.60 | -0.09 | 0.20 | 0.00 | 0.08 | 0.08 | 0.05 |
| | TracIn | -0.68 | -0.81 | -0.64 | -0.94 | -1.00 | -1.00 | -0.11 | -0.12 | -0.09 |
| | TRAK | 0.43 | 0.36 | 0.29 | -0.01 | 0.20 | 0.00 | -0.15 | -0.14 | -0.10 |
| | Ours | **0.70** | **0.87** | **0.64** | **1.00** | **1.00** | **1.00** | **0.11** | **0.14** | **0.10** |
| | DAVINZ | -0.67 | -0.71 | -0.49 | 0.88 | 0.80 | 0.67 | 0.04 | 0.06 | 0.04 |
| | Deviation | 0.32 | 0.20 | 0.13 | -0.80 | -0.80 | -0.67 | 0.14 | 0.15 | 0.10 |
| Qwen3-0.6B | LOGRA | -0.62 | -0.77 | -0.58 | -0.87 | -0.80 | -0.67 | -0.07 | -0.04 | -0.03 |
| | TracIn | -0.69 | -0.66 | -0.49 | -0.87 | -0.80 | -0.67 | 0.05 | 0.06 | 0.04 |
| | TRAK | 0.64 | **0.73** | 0.54 | -0.91 | -0.80 | -0.67 | -0.01 | -0.02 | -0.02 |
| | Ours | **0.86** | 0.71 | **0.63** | **1.00** | **1.00** | **1.00** | **0.25** | **0.26** | **0.18** |
| | DAVINZ | -0.41 | -0.32 | -0.33 | 0.66 | 0.20 | 0.00 | 0.43 | **0.41** | **0.30** |
| | Deviation | -0.23 | -0.52 | -0.42 | -0.87 | -0.80 | -0.67 | -0.10 | -0.09 | -0.07 |
| Qwen3-1.7B | LOGRA | 0.19 | 0.24 | 0.11 | -0.60 | -0.20 | 0.00 | -0.41 | -0.40 | -0.30 |
| | TracIn | -0.03 | 0.02 | 0.07 | -0.63 | -0.60 | -0.33 | 0.22 | 0.34 | 0.24 |
| | TRAK | 0.40 | 0.32 | 0.33 | -0.60 | -0.20 | 0.00 | -0.17 | -0.18 | -0.13 |
| | Ours | **0.70** | **0.81** | **0.69** | **1.00** | **1.00** | **1.00** | **0.44** | **0.41** | **0.30** |
| | DAVINZ | 0.70 | 0.62 | 0.45 | 0.14 | **0.80** | **0.67** | -0.11 | -0.06 | -0.04 |
| | Deviation | 0.59 | 0.79 | 0.58 | -0.05 | -0.40 | -0.33 | 0.16 | 0.15 | 0.10 |
| Llama-3.2-1B-Instruct | LOGRA | 0.51 | 0.65 | 0.49 | -0.07 | -0.80 | -0.67 | -0.06 | 0.06 | 0.05 |
| | TracIn | 0.63 | 0.45 | 0.36 | **0.27** | 0.00 | 0.00 | -0.04 | -0.18 | -0.12 |
| | TRAK | -0.53 | -0.38 | -0.27 | -0.25 | -0.80 | -0.67 | -0.01 | -0.03 | -0.02 |
| | Ours | **0.96** | **0.84** | **0.72** | -0.21 | -0.80 | -0.67 | **0.24** | **0.27** | **0.19** |

## 5.3 EFFECTIVE AND EFFICIENT DATA VALUATION

We compare our method against five recent baselines across the four tasks, including image classification, sentiment classification, instruction following, and complex reasoning. Effectiveness is measured by correlations between data valuation scores and ground-truth including Pearson, Spearman, and Kendall correlations, and efficiency is assessed by runtime.

From Figure 7, Figure 8, and Table 1, we have the following observations: (1) Across most tasks and backbones, our method achieves the highest correlation scores with ground-truth performance, and these gains are consistent across correlation measures, demonstrating its effectiveness in identifying valuable data contributors. In particular, on the sentiment task with Qwen3-1.7B, our approach attains a Spearman correlation of 0.81, significantly exceeding the second-best method of 0.32. (2) In addition to its effectiveness, our method incurs a low runtime, requiring only 8 seconds on average. This is significantly faster than Retrain (627 seconds), Deviation (553 seconds), and TRAK (166 seconds), highlighting better computational efficiency.

Table 2: Relative scores of our method's NTK and MMD components of selected data contributors under different training sizes. Scores are min-max normalized across contributors within each subsampling size to highlight relative rankings.

| Size | Contributor 1 | | Contributor 2 | | Contributor 3 | | Contributor 4 | | Contributor 5 | |
|---|---|---|---|---|---|---|---|---|---|---|
| | MMD | NTK | MMD | NTK | MMD | NTK | MMD | NTK | MMD | NTK |
| 100 | 0.00 | 0.00 | 0.18 | 0.55 | 0.00 | 0.33 | 0.38 | 0.96 | 1.00 | 1.00 |
| 400 | 0.00 | 0.00 | 0.17 | 0.53 | 0.03 | 0.36 | 0.20 | 0.74 | 1.00 | 1.00 |
| 1,000 | 0.00 | 0.00 | 0.15 | 0.66 | 0.03 | 0.54 | 0.32 | 0.91 | 1.00 | 1.00 |
| 4,000 | 0.00 | 0.00 | 0.23 | 0.65 | 0.06 | 0.53 | 0.29 | 0.92 | 1.00 | 1.00 |

## 5.4 Stability of Relative Valuation Under Subsampling

To examine the stability of our method under subsampled training sets, we analyze whether MMD score $\pi \text{Dist}(\boldsymbol{T}, \boldsymbol{S}_1) + (1-\pi)\text{Dist}(\boldsymbol{T}, \boldsymbol{S}_2)$ and NTK score $\sqrt{\hat{\boldsymbol{y}}^\top \boldsymbol{\Theta}_0^{-1} \hat{\boldsymbol{y}} / |\boldsymbol{S}|}$ in Eq.(8) remain stable when computed on a small fraction of the data. Specifically, we conduct an image classification task with the first five contributors, and compute their scores using training subsets of size 100, 400, 1,000, and 4,000. For comparability, we apply min-max normalization to the scores within each subsampling size, focusing on the relative rankings rather than absolute values.

As shown in Table 2, both MMD and NTK scores maintain stability in their relative contributor rankings across different subsample sizes. For example, contributor 1 consistently receives the lowest normalized MMD and NTK scores, while contributor 5 consistently receives the highest score, regardless of training size. This suggests that the relative quality of contributors given by our method is consistent across diverse subsampling sizes.

## 6 Related Work

**Data Valuation.** Data valuation methods quantify the contribution or importance of individual data subsets of a dataset to the performance of machine learning models. Traditional retraining-based approaches, such as LOO (Koh et al., 2019; Koh & Liang, 2017), SV-based methods (Ghorbani & Zou, 2019), and downsampling Yoon et al. (2020), require extensive computation due to model retraining, making them infeasible for LLMs. Recently, gradient-based methods emerged as efficient alternatives, leveraging model gradients and checkpoints for data valuation. TracIn (Pruthi et al., 2020) specifically traces the gradient descent path of training, estimating influence based on gradient similarity across training checkpoints. TRAK (Park et al., 2023) approximates the influence using kernel methods derived from gradients and efficient random projections, scaling effectively to large-scale models and datasets. DAVINZ (Wu et al., 2022) leverages the NTK to estimate data valuation directly from initialization gradients, enabling a training-free evaluation. LOGRA (Choe et al., 2024) introduces a label-only gradient attribution approach, estimating data valuation by analyzing gradient alignment without relying on explicit labels. Despite these advances, current methods still face significant limitations when dealing with datasets composed of real and synthetic data.

**LLM Model Collapse.** LLMs trained with increasing amounts of synthetic data have been observed to suffer from model collapse, a phenomenon where model performance degrades over training (Shumailov et al., 2024; Dohmatob et al., 2024b). One key cause is synthetic data often exhibits reduced diversity and redundancy in knowledge compared to real data, especially when generated from earlier versions of the same model (Havrilla et al., 2024; Chen et al., 2024a). As synthetic data are reused or recursively generated, the information content becomes increasingly narrow and biased, resulting in amplified errors (Shumailov et al., 2023; Zhang et al., 2024a). These issues motivate a principled understanding of the LLM training behaviors on datasets of real and synthetic mixtures.

## 7 Conclusion

LLMs trained on datasets composed of real and synthetic mixtures exhibit complex scaling behaviors. In this work, we identify a fine-grained three-phase scaling behavior with two breakpoints, reflecting transitions in the model's ability to acquire head and tail knowledge. We further derive a general LLM generalization bound to reveal key factors that influence the performance of LLMs. Building on this theoretical bound, we develop a practical data valuation method that estimates the contribution of individual data subsets. Empirical results on four diverse tasks show that our method achieves higher correlation with ground-truth than baseline methods, while remaining computationally efficient at LLM-scale tasks.

## REPRODUCIBILITY STATEMENT

We take several steps to ensure that our work is fully reproducible. In particular, we provide proofs of the theoretical results in Appendix A. In addition, we provide a detailed explanation of our method in Section 4, along with pseudo code in Algorithm 1. The experimental details, including the choice of hyperparameters and data preprocessing, are outlined in detail in Section 5.1 and Appendix appendix D. An anonymized version of the code used to reproduce our results can be found at `https://anonymous.4open.science/r/3phaseLLM-E5E2/`. All datasets used in our experiments are publicly accessible.

## ETHICS STATEMENT

Our work does not involve human or animal subjects, personally identifiable data, or sensitive information. The datasets used are publicly available, and we follow their respective licenses. The methods and findings presented do not pose foreseeable risks of misuse, discrimination, or harm. We therefore believe our work raises no specific ethical concerns under the ICLR Code of Ethics.

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

# A    Proofs of Theoretical Analysis

This section provides complete proofs for the theoretical analysis. We first give the definition of distribution discrepancy (Gretton et al., 2012a) between $\mathcal{D}_T$ and $\mathcal{D}_S$ as a measure to quantify distribution divergence in Definition 1.

**Definition 1.** *Given any function space $\mathcal{H}$, the distribution discrepancy between $\mathcal{D}_T$ and $\mathcal{D}_S$ is defined as:*

$$d_{\mathcal{H}}(\mathcal{D}_T, \mathcal{D}_S) \triangleq \sup_{h \in \mathcal{H}} \left| \mathbb{E}_{\boldsymbol{x}' \sim \mathcal{D}_T}[h(\boldsymbol{x}')] - \mathbb{E}_{\boldsymbol{x} \sim \mathcal{D}_S}[h(\boldsymbol{x})] \right|,$$

*which can be empirically estimated using samples $\boldsymbol{S}$ and $\boldsymbol{T}$ from the respective $\mathcal{D}_S$ and $\mathcal{D}_T$:*

$$d_{\mathcal{H}}(\boldsymbol{T}, \boldsymbol{S}) \triangleq \sup_{h \in \mathcal{H}} \left| \frac{1}{|\boldsymbol{T}|} \sum_{i=1}^{|\boldsymbol{T}|} h(\boldsymbol{x}'_i) - \frac{1}{|\boldsymbol{S}|} \sum_{i=1}^{|\boldsymbol{S}|} h(\boldsymbol{x}_i) \right|.$$

We then introduce the following lemma, which is adapted from the proof of Theorem 1 in (Wu et al., 2022) and the proof of Theorem 2 in (Shu et al., 2022).

**Lemma 2.** *Assume that $\lambda_{\min}(\boldsymbol{\Theta}_0) > 0$ and $||\nabla_{\boldsymbol{\theta}} f(\boldsymbol{x}; \boldsymbol{\theta}_0)||_2 \leq B$ for any $(\boldsymbol{x}, y) \in \boldsymbol{S}$ sampled from $\mathcal{D}_S$ with $||\boldsymbol{x}||_2 \leq 1$ and $y \in [0, 1]$. Given the loss function $\ell(f, y) \triangleq (f - y)^2/2$ and define $\hat{\boldsymbol{y}} \triangleq y - f(\boldsymbol{x})$, there exist constants $c > 0$ and $M \in \mathbb{N}$ such that for every $m > M$, when applying gradient descent with learning rate*

$$\eta < \min \left\{ 2m^{-1} \left( \lambda_{\min}(\boldsymbol{\Theta}_\infty) + \lambda_{\max}(\boldsymbol{\Theta}_\infty) \right)^{-1}, \ |\boldsymbol{S}| \lambda_{\max}^{-1}(\boldsymbol{\Theta}_0) \right\},$$

*for all the functions $f_t$ obtained during the optimization, with high probability $(1 - \delta)$ over the dataset $\boldsymbol{S}$ of size $|\boldsymbol{S}|$, we have*

$$\mathcal{L}_{\mathcal{D}_S}(f_t) \leq \mathcal{L}_S(f_t) + 2B\sqrt{\hat{\boldsymbol{y}}^\top \boldsymbol{\Theta}_0^{-1} \hat{\boldsymbol{y}} / |\boldsymbol{S}|} + \varepsilon,$$

*where $\hat{\boldsymbol{y}} = [\hat{y}_1, \ldots, \hat{y}_{|\boldsymbol{S}|}]^\top$, $\varepsilon \triangleq 2c/\sqrt{m} + 3\sqrt{\log(4/\delta)/2|\boldsymbol{S}|}$, and $\lambda_{\min}(\cdot), \lambda_{\max}(\cdot)$ denote the minimum and maximum eigenvalue of a matrix, respectively.*

With the above definition and lemma, we are now ready to prove Theorem 1.

**Theorem 1** (LLM Generalization Bound under Real and Synthetic Mixtures). *Let $\lambda_{\min}(\cdot)$ and $\lambda_{\max}(\cdot)$ denote the minimum and maximum eigenvalue of a matrix. Assume $\lambda_{\min}(\boldsymbol{\Theta}_0) > 0$ and $||\nabla_{\boldsymbol{\theta}} f(\boldsymbol{x}; \boldsymbol{\theta}_0)||_2 \leq B$ for any $(\boldsymbol{x}, y) \in \boldsymbol{S}$ with $||\boldsymbol{x}||_2, y \in [0, 1]$. There exist $M \in \mathbb{N}$ such that for every $m > M$, when applying gradient descent with learning rate $\eta < \min \left\{ 2m^{-1} \left( \lambda_{\min}(\boldsymbol{\Theta}_\infty) + \lambda_{\max}(\boldsymbol{\Theta}_\infty) \right)^{-1}, |\boldsymbol{S}|/\lambda_{\max}(\boldsymbol{\Theta}_0) \right\}$, with probability at least $1 - 2\delta$,*

$$\mathcal{L}_{D_T}(f) \leq \pi \mathcal{L}_{\boldsymbol{S}_1}(f) + (1 - \pi)\mathcal{L}_{\boldsymbol{S}_2}(f) + \pi d_{\mathcal{H}}(\boldsymbol{T}, \boldsymbol{S}_1) + (1 - \pi)d_{\mathcal{H}}(\boldsymbol{T}, \boldsymbol{S}_2)$$
$$+ 2B\sqrt{\frac{\hat{\boldsymbol{y}}^T \boldsymbol{\Theta}_0^{-1} \hat{\boldsymbol{y}}}{|\boldsymbol{S}|}} + \sqrt{\frac{2\max(\pi, 1 - \pi)\log(8/\delta)}{|\boldsymbol{S}|}} + \varepsilon, \quad (7)$$

*where each element in $\hat{\boldsymbol{y}}$ is defined as $\hat{y} \triangleq y - f(\boldsymbol{x}; \boldsymbol{\theta}_0)$ and $\varepsilon \triangleq 2c/\sqrt{m} + 3\sqrt{\log(4/\delta)/2|\boldsymbol{S}|} + \sqrt{\log(4/\delta)/2|\boldsymbol{T}|} + \mathcal{L}_{\mathcal{D}_T}(f^*) + \mathcal{L}_{\mathcal{D}_S}(f^*)$, and $c > 0$ is a constant.*

*Proof.* Let $\phi_S$ and $\phi_T$ be the probability density function for data distribution $\mathcal{D}_S$ and $\mathcal{D}_T$, respectively. From (Ben-David et al., 2010), the generalization performance on $\mathcal{D}_T$ can therefore be bounded using the generalization performance on $\mathcal{D}_S$ by assuming that the loss function $\ell(\cdot, \cdot)$ is

$\mu$-Lipschitz continuous, where $\mu > 0$ denotes a Lipschitz constant:

$$
\begin{aligned}
\mathcal{L}_{\mathcal{D}_T}(f) \leq & \mathcal{L}_{\mathcal{D}_T}(f^*) + \mathbb{E}_{(\boldsymbol{x},y) \sim \mathcal{D}_T} |\ell(f(\boldsymbol{x}),y) - \ell(f^*(\boldsymbol{x}),y)| \\
\leq & \mathcal{L}_{\mathcal{D}_T}(f^*) + \mathbb{E}_{(\boldsymbol{x},y) \sim \mathcal{D}_S} |\ell(f(\boldsymbol{x}),y) - \ell(f^*(\boldsymbol{x}),y)| + \\
& \left| \mathbb{E}_{(\boldsymbol{x},y) \sim \mathcal{D}_S} |\ell(f(\boldsymbol{x}),y) - \ell(f^*(\boldsymbol{x}),y)| - \mathbb{E}_{(\boldsymbol{x},y) \sim \mathcal{D}_T} |\ell(f(\boldsymbol{x}),y) - \ell(f^*(\boldsymbol{x}),y)| \right| \\
\leq & \mathcal{L}_{\mathcal{D}_T}(f^*) + \mathbb{E}_{(\boldsymbol{x},y) \sim \mathcal{D}_S} \left[ |\ell(f(\boldsymbol{x}),y) - \ell(f^*(\boldsymbol{x}),y)| \right] + \\
& \left| \int (\phi_S(\boldsymbol{x}) - \phi_T(\boldsymbol{x})) \left( \ell(f(\boldsymbol{x}),y) - \ell(f^*(\boldsymbol{x}),y) \right) d\boldsymbol{x} \right| \\
\leq & \mathcal{L}_{\mathcal{D}_T}(f^*) + \mathbb{E}_{(\boldsymbol{x},y) \sim \mathcal{D}_S} \left( \ell(f(\boldsymbol{x}),y) + \ell(f^*(\boldsymbol{x}),y) \right) + \\
& \mu \left| \int (\phi_S(\boldsymbol{x}) - \phi_T(\boldsymbol{x})) |f(\boldsymbol{x}) - f^*(\boldsymbol{x})| d\boldsymbol{x} \right| \\
\leq & \mathcal{L}_{\mathcal{D}_T}(f^*) + \mathbb{E}_{(\boldsymbol{x},y) \sim \mathcal{D}_S} \ell(f(\boldsymbol{x}),y) + \mathbb{E}_{(\boldsymbol{x},y) \sim \mathcal{D}_S} \ell(f^*(\boldsymbol{x}),y) + \\
& \mu \left| \int (\phi_S(\boldsymbol{x}) - \phi_T(\boldsymbol{x})) h(\boldsymbol{x}) d\boldsymbol{x} \right| \\
\leq & \mathcal{L}_{\mathcal{D}_T}(f^*) + \mathcal{L}_{\mathcal{D}_S}(f^*) + \mathcal{L}_{\mathcal{D}_S}(f) + \mu \sup_{h \in \mathcal{H}} |\mathbb{E}_{\mathcal{D}_S}[h(\boldsymbol{x})] - \mathbb{E}_{\mathcal{D}_T}[h(\boldsymbol{x})]| \\
\leq & \mathcal{L}_{\mathcal{D}_T}(f^*) + \mathcal{L}_{\mathcal{D}_S}(f^*) + \mathcal{L}_{\mathcal{D}_S}(f) + \mu d_{\mathcal{H}}(\mathcal{D}_S, \mathcal{D}_T).
\end{aligned}
\tag{9}
$$

Next, we approximate $d_{\mathcal{H}}(\mathcal{D}_S, \mathcal{D}_T)$ using $d_{\mathcal{H}}(\boldsymbol{T}, \boldsymbol{S}_1)$ and $d_{\mathcal{H}}(\boldsymbol{T}, \boldsymbol{S}_2)$ where $\boldsymbol{T}$, $\boldsymbol{S}_1$, and $\boldsymbol{S}_2$ denote the test, real and synthetic datasets. Following Hoeffding's inequality and the assumption stated in the main text that $h(\boldsymbol{x}) \leq 1$, we have:

$$
\begin{aligned}
& \mathbb{P}\left( \left| \mathbb{E}_{\mathcal{D}_S}[h(\boldsymbol{x})] - \frac{1}{|\boldsymbol{S}|} \sum_{i=1}^{|\boldsymbol{S}|} h(\boldsymbol{x}_i) \right| \geq \varepsilon \right) \\
\leq & \mathbb{P}\left( \left| \mathbb{E}_{\mathcal{D}_S}[h(\boldsymbol{x})] - \frac{1}{\pi|\boldsymbol{S}|} \sum_{i=1}^{\pi|\boldsymbol{S}|} h(\boldsymbol{x}_i) \right| \geq \frac{\varepsilon}{2\pi} \right) + \\
& \mathbb{P}\left( \left| \mathbb{E}_{\mathcal{D}_S}[h(\boldsymbol{x})] - \frac{1}{(1-\pi)|\boldsymbol{S}|} \sum_{i=\pi|\boldsymbol{S}|+1}^{|\boldsymbol{S}|} h(\boldsymbol{x}_i) \right| \geq \frac{\varepsilon}{2(1-\pi)} \right) \\
\leq & 2\exp\left( -\frac{\varepsilon^2 |\boldsymbol{S}|}{2\pi} \right) + 2\exp\left( -\frac{\varepsilon^2 |\boldsymbol{S}|}{2(1-\pi)} \right) \\
\leq & 4 \max\left\{ \exp\left( -\frac{\varepsilon^2 |\boldsymbol{S}|}{2\pi} \right), \exp\left( -\frac{\varepsilon^2 |\boldsymbol{S}|}{2(1-\pi)} \right) \right\} \\
= & 4 \exp\left( -\frac{\varepsilon^2 |\boldsymbol{S}|}{2 \max(\pi, 1-\pi)} \right).
\end{aligned}
\tag{10}
$$

Then the following inequality holds with probability at least $1 - \delta$:

$$
\begin{aligned}
& |\mathbb{E}_{\mathcal{D}_S}[h(\boldsymbol{x})] - \mathbb{E}_{\mathcal{D}_T}[h(\boldsymbol{x})]| - \left| \frac{1}{|\boldsymbol{S}|} \sum_{i=1}^{|\boldsymbol{S}|} h(\boldsymbol{x}_i) - \frac{1}{|\boldsymbol{T}|} \sum_{i=1}^{|\boldsymbol{T}|} h(\boldsymbol{x}_i') \right| \\
\leq & \left| \mathbb{E}_{\mathcal{D}_S}[h(\boldsymbol{x})] - \frac{1}{|\boldsymbol{S}|} \sum_{i=1}^{|\boldsymbol{S}|} h(\boldsymbol{x}_i) \right| + \left| \mathbb{E}_{\mathcal{D}_T}[h(\boldsymbol{x})] - \frac{1}{|\boldsymbol{T}|} \sum_{i=1}^{|\boldsymbol{T}|} h(\boldsymbol{x}_i') \right| \\
\leq & \sqrt{\frac{2 \max(\pi, 1-\pi) \log(4/\delta)}{|\boldsymbol{S}|}} + \sqrt{\frac{\log(4/\delta)}{2|\boldsymbol{T}|}}.
\end{aligned}
\tag{11}
$$

Based on the inequality above, we can approximate $d_{\mathcal{H}}(\mathcal{D}_S, \mathcal{D}_T)$ using $d_{\mathcal{H}}(\boldsymbol{T}, \boldsymbol{S}_1)$ and $d_{\mathcal{H}}(\boldsymbol{T}, \boldsymbol{S}_2)$ as below with probability at least $1 - \delta$:

$$d_{\mathcal{H}}(\mathcal{D}_S, \mathcal{D}_T) = \sup_{h \in \mathcal{H}} |\mathbb{E}_{\mathcal{D}_S}[h(\boldsymbol{x})] - \mathbb{E}_{\mathcal{D}_T}[h(\boldsymbol{x})]|$$

$$\leq \sup_{h \in \mathcal{H}} \left| \frac{1}{|\boldsymbol{S}|} \sum_{i=1}^{|\boldsymbol{S}|} h(\boldsymbol{x}_i) - \frac{1}{|\boldsymbol{T}|} \sum_{i=1}^{|\boldsymbol{T}|} h(\boldsymbol{x}_i') \right| +$$

$$\sqrt{\frac{2 \max(\pi, 1-\pi) \log(4/\delta)}{|\boldsymbol{S}|}} + \sqrt{\frac{\log(4/\delta)}{2|\boldsymbol{T}|}}$$

$$\leq \pi \sup_{h \in \mathcal{H}} \left| \frac{1}{|\boldsymbol{T}|} \sum_{i=1}^{|\boldsymbol{T}|} h(\boldsymbol{x}_i') - \frac{1}{\pi|\boldsymbol{S}|} \sum_{i=1}^{\pi|\boldsymbol{S}|} h(\boldsymbol{x}_i) \right| +$$

$$(1-\pi) \sup_{h \in \mathcal{H}} \left| \frac{1}{|\boldsymbol{T}|} \sum_{i=1}^{|\boldsymbol{T}|} h(\boldsymbol{x}_i') - \frac{1}{(1-\pi)|\boldsymbol{S}|} \sum_{i=\pi|\boldsymbol{S}|+1}^{|\boldsymbol{S}|} h(\boldsymbol{x}_i) \right| +$$

$$\sqrt{\frac{2 \max(\pi, 1-\pi) \log(4/\delta)}{|\boldsymbol{S}|}} + \sqrt{\frac{\log(4/\delta)}{2|\boldsymbol{T}|}}$$

$$\leq \pi d_{\mathcal{H}}(\boldsymbol{T}, \boldsymbol{S}_1) + (1-\pi) d_{\mathcal{H}}(\boldsymbol{T}, \boldsymbol{S}_2) + \sqrt{\frac{2 \max(\pi, 1-\pi) \log(4/\delta)}{|\boldsymbol{S}|}} + \sqrt{\frac{\log(4/\delta)}{2|\boldsymbol{T}|}}. \tag{12}$$

For the empirical loss, we have:

$$L_S(f) = \frac{1}{|\boldsymbol{S}|} \sum_{i=1}^{|\boldsymbol{S}|} \ell(f(\boldsymbol{x}_i), y_i)$$

$$= \frac{\pi}{\pi|\boldsymbol{S}|} \sum_{i=1}^{\pi|\boldsymbol{S}|} \ell(f(\boldsymbol{x}_i), y_i) + \frac{1-\pi}{(1-\pi)|\boldsymbol{S}|} \sum_{i=\pi|\boldsymbol{S}|+1}^{|\boldsymbol{S}|} \ell(f(\boldsymbol{x}_i), y_i) \tag{13}$$

$$= \pi L_{\boldsymbol{S}_1}(f) + (1-\pi) L_{\boldsymbol{S}_2}(f).$$

Note that $\mu = 1$ for loss function $\ell(f, y) \triangleq (f - y)^2/2$ when $f, y \in [0, 1]$. By combining the results in Eq.(9), Eq.(12) and Eq.(13), and integrating the conclusion in Lemma 2, we complete the proof. $\qquad \square$

Theorem 1 provides a general theoretical understanding of LLMs trained on mixtures of real and synthetic data. Building on this foundation, we next reveal a three-phase transition in the scaling behavior of LLMs under certain assumptions on data and model in Lemma 1.

**Lemma 1** (Scaling Behavior with Three phases). *Consider training data where the probability of knowledge $i$ is $q_i = \pi p_i + (1 - \pi)p_i'$, where $p_i \propto i^{-\beta}$ and $p_i'$ is cut off at rank $k$ as defined above. The test error $\mathcal{L}_{test}$ exhibits distinct scaling regimes characterized by two breakpoints at sample sizes $|\boldsymbol{S}| = k^\beta$ and $|\boldsymbol{S}| = k^\beta/\pi$. We have[1]:*
*Phase 1 (Rapid-Learning): $|\boldsymbol{S}| \leq c_1 k^\beta$, where $c_1$ is absolute constant,*

$$\mathcal{L}_{test} \asymp a |\boldsymbol{S}|^{\frac{1-\alpha-\beta}{\beta}} - b |\boldsymbol{S}|^{\frac{1-\lambda-\beta}{\beta}} + a k^{1-\alpha-\beta} - b k^{1-\lambda-\beta} + k^{1-\beta}. \tag{5}$$

*Phase 2 (Plateau): $c_1 k^\beta < |\boldsymbol{S}| < c_2 k^\beta/\pi$, where $c_2$ is absolute constant, $\mathcal{L}_{test}$ enters a transition state as the limited presence of tail knowledge prevents the rapid learning.*
*Phase 3 (Tail-Learning): $|\boldsymbol{S}| \geq c_2 k^\beta/\pi$,*

$$\mathcal{L}_{test} \asymp a(\pi|\boldsymbol{S}|)^{\frac{1-\alpha-\beta}{\beta}} - b(\pi|\boldsymbol{S}|)^{\frac{1-\lambda-\beta}{\beta}} + k^{1-\beta}. \tag{6}$$

---

[1]The notation $g(n) \asymp h(n)$ means that $c_1 h(n) \leq g(n) \leq c_2 h(n)$ for sufficiently large $n$ and absolute constants $c_1, c_2 > 0$.

*Proof.* From Eq. (9) and triangle inequality, we have

$$
\begin{aligned}
\mathcal{L}_{\mathcal{D}_T}(f) &\leq \mathcal{L}_{\mathcal{D}_S}(f) + d_{\mathcal{H}}\left(\mathcal{D}_S, \mathcal{D}_T\right) + \mathcal{L}_{\mathcal{D}_T}(f^*) + \mathcal{L}_{\mathcal{D}_S}(f^*) \\
&\leq \mathcal{L}_{\mathcal{D}}(f) + d_{\mathcal{H}}\left(\mathcal{D}_S, \mathcal{D}\right) + d_{\mathcal{H}}\left(\mathcal{D}_S, \mathcal{D}_T\right) + \mathcal{L}_{\mathcal{D}_T}(f^*) + 2\mathcal{L}_{\mathcal{D}_S}(f^*) + \mathcal{L}_{\mathcal{D}}(f^*) \\
&\leq \mathcal{L}_{\mathcal{D}}(f) + d_{\mathcal{H}}\left(\mathcal{D}, \mathcal{D}_T\right) + 2d_{\mathcal{H}}\left(\mathcal{D}_S, \mathcal{D}\right) + \mathcal{L}_{\mathcal{D}_T}(f^*) + 2\mathcal{L}_{\mathcal{D}_S}(f^*) + \mathcal{L}_{\mathcal{D}}(f^*),
\end{aligned}
\tag{14}
$$

where $\mathcal{D}$ is the true distribution.

We also have

$$
\mathcal{L}_{\mathcal{D}}(f) \leq \mathcal{L}_{\mathcal{D}_T}(f) + d_{\mathcal{H}}\left(\mathcal{D}_T, \mathcal{D}\right) + \mathcal{L}_{\mathcal{D}_T}(f^*) + \mathcal{L}_{\mathcal{D}}(f^*).
\tag{15}
$$

Suppose $d_{\mathcal{H}}\left(\mathcal{D}, \mathcal{D}_T\right) = 0$ as $\mathcal{D}$ and $\mathcal{D}_T$ follow the same distribution, then we have

$$
\begin{aligned}
\mathcal{L}_{\mathcal{D}_T}(f) &= \mathbb{E}_{(\boldsymbol{x},y)\sim\mathcal{D}_T}[\ell(f(\boldsymbol{x}),y)] \\
&\asymp \mathcal{L}_{\mathcal{D}}(f) + d_{\mathcal{H}}\left(\mathcal{D}_S, \mathcal{D}\right) = \mathbb{E}_{(\boldsymbol{x},y)\sim\mathcal{D}}[\ell(f(\boldsymbol{x}),y)] + d_{\mathcal{H}}\left(\mathcal{D}_S, \mathcal{D}\right)
\end{aligned}
\tag{16}
$$

for any $f$ gained based on the training dataset $\boldsymbol{S} \sim \mathcal{D}_S$.

Calculating expectation on $\mathcal{D}_S$, we have:

$$
\begin{aligned}
E_{\text{test}} &= \mathbb{E}_{\mathcal{D}_S}\left[\mathbb{E}_{(\boldsymbol{x},y)\sim\mathcal{D}_T}[\ell(f(\boldsymbol{x}),y)]\right] \\
&\asymp \mathbb{E}_{\mathcal{D}_S}\left[\mathbb{E}_{(\boldsymbol{x},y)\sim\mathcal{D}}[\ell(f(\boldsymbol{x}),y)]\right] + \mathbb{E}_{\mathcal{D}_S}\left[d_{\mathcal{H}}\left(\mathcal{D}_S, \mathcal{D}\right)\right].
\end{aligned}
\tag{17}
$$

For the first term,

$$
\begin{aligned}
&\mathbb{E}_{\mathcal{D}_S}\left[\mathbb{E}_{(\boldsymbol{x},y)\sim\mathcal{D}}[\ell(f(\boldsymbol{x}),y)]\right] \\
&\asymp \sum_{i\geq 1} p_i\left[(1-(1-q_i)^{|\boldsymbol{S}|})(1-\rho(i)) + (1-q_i)^{|\boldsymbol{S}|}(1-\gamma(i))\right] \\
&\asymp \sum_{i\geq 1} p_i(1-\rho(i)) + \sum_{1\leq i\leq k} p_i(\rho(i)-\gamma(i))(1-p_i)^{|\boldsymbol{S}|} + \sum_{i\geq k+1} p_i(\rho(i)-\gamma(i))(1-\pi p_i)^{|\boldsymbol{S}|} \\
&\asymp \frac{1}{\beta-1} - \frac{a}{\alpha+\beta-1} + \frac{a}{\beta}|\boldsymbol{S}|^{\frac{1-\alpha-\beta}{\beta}}\left[\Gamma\left(\frac{\alpha+\beta-1}{\beta},|\boldsymbol{S}|k^{-\beta}\right) - \Gamma\left(\frac{\alpha+\beta-1}{\beta},|\boldsymbol{S}|\right)\right] - \\
&\quad \frac{b}{\beta}|\boldsymbol{S}|^{\frac{1-\lambda-\beta}{\beta}}\left[\Gamma\left(\frac{\lambda+\beta-1}{\beta},|\boldsymbol{S}|k^{-\beta}\right) - \Gamma\left(\frac{\lambda+\beta-1}{\beta},|\boldsymbol{S}|\right)\right] - \\
&\quad \frac{a}{\beta}(\pi|\boldsymbol{S}|)^{\frac{1-\alpha-\beta}{\beta}}\Gamma\left(\frac{\alpha+\beta-1}{\beta},\pi|\boldsymbol{S}|(k+1)^{-\beta}\right) + \\
&\quad \frac{b}{\beta}(\pi|\boldsymbol{S}|)^{\frac{1-\lambda-\beta}{\beta}}\Gamma\left(\frac{\lambda+\beta-1}{\beta},\pi|\boldsymbol{S}|(k+1)^{-\beta}\right),
\end{aligned}
\tag{18}
$$

where $\Gamma(s,x) = \int_x^{\infty} t^{s-1}e^{-t}\, dt$ is the upper incomplete gamma function.

When $|\boldsymbol{S}| \leq c_1 k^{\beta}$, where $c_1$ is a constant, we have $\Gamma\left(\frac{\alpha+\beta-1}{\beta},|\boldsymbol{S}|k^{-\beta}\right) - \Gamma\left(\frac{\alpha+\beta-1}{\beta},|\boldsymbol{S}|\right) = \Theta(1) - o(1) = \Theta(1)$; when $|\boldsymbol{S}| > c_1 k^{\beta}$, we have $\Gamma\left(\frac{\alpha+\beta-1}{\beta},|\boldsymbol{S}|k^{-\beta}\right) - \Gamma\left(\frac{\alpha+\beta-1}{\beta},|\boldsymbol{S}|\right) = o(1) - o(1) = o(1)$. Similarly, when $|\boldsymbol{S}| \leq c_1 k^{\beta}$, where $c_1$ is a constant, we have $\Gamma\left(\frac{\lambda+\beta-1}{\beta},|\boldsymbol{S}|k^{-\beta}\right) - \Gamma\left(\frac{\lambda+\beta-1}{\beta},|\boldsymbol{S}|\right) = \Theta(1) - o(1) = \Theta(1)$; when $|\boldsymbol{S}| > c_1 k^{\beta}$, we have $\Gamma\left(\frac{\lambda+\beta-1}{\beta},|\boldsymbol{S}|k^{-\beta}\right) - \Gamma\left(\frac{\lambda+\beta-1}{\beta},|\boldsymbol{S}|\right) = o(1) - o(1) = o(1)$. The test loss for knowledge 1 to $k$ is related to $\sum_{1\leq i\leq k} p_i\left[(1-(1-q_i)^{|\boldsymbol{S}|})(1-\rho(i)) + (1-q_i)^{|\boldsymbol{S}|}(1-\gamma(i))\right]$, thus the breakpoint for head knowledge is $|\boldsymbol{S}| = c_1 k^{\beta}$. When $\pi|\boldsymbol{S}| \geq c_2 k^{\beta}$, where $c_2$ is a constant, we have $\Gamma\left(\frac{\alpha+\beta-1}{\beta},\pi|\boldsymbol{S}|(k+1)^{-\beta}\right) = \Theta(1)$; when $\pi|\boldsymbol{S}| < c_2 k^{\beta}$, we have $\Gamma\left(\frac{\alpha+\beta-1}{\beta},\pi|\boldsymbol{S}|(k+1)^{-\beta}\right) = \frac{\beta}{1-\alpha-\beta}\Theta((\pi|\boldsymbol{S}|k^{-\beta})^{\frac{\alpha+\beta-1}{\beta}})$. Similarly, when $\pi|\boldsymbol{S}| \geq c_2 k^{\beta}$, where $c_2$ is a constant, we have $\Gamma\left(\frac{\lambda+\beta-1}{\beta},\pi|\boldsymbol{S}|(k+1)^{-\beta}\right) = \Theta(1)$; when $\pi|\boldsymbol{S}| < c_2 k^{\beta}$, we have $\Gamma\left(\frac{\lambda+\beta-1}{\beta},\pi|\boldsymbol{S}|(k+1)^{-\beta}\right) = \frac{\beta}{1-\lambda-\beta}\Theta((\pi|\boldsymbol{S}|k^{-\beta})^{\frac{\lambda+\beta-1}{\beta}})$. The test loss for knowledge beyond

rank $k$ is related to $\sum_{i \geq k+1} p_i \left[ (1 - (1 - q_i)^{|\boldsymbol{S}|})(1 - \rho(i)) + (1 - q_i)^{|\boldsymbol{S}|}(1 - \gamma(i)) \right]$, thus the breakpoint for tail knowledge is $|\boldsymbol{S}| = c_2 k^\beta / \pi$.

For the second term,

$$\mathbb{E}_{\mathcal{D}_S}\left[ d_{\mathcal{H}}\left( \mathcal{D}_S, \mathcal{D} \right) \right] \asymp \sum_{i=k+1}^{\infty} p_i = \int_{k+1}^{\infty} x^{-\beta} dx = \frac{(k+1)^{1-\beta}}{\beta - 1}. \tag{19}$$

By combining the results above, we complete the proof. □

## B  PSEUDO CODE

This section presents the pseudo code for the proposed data valuation method in Algorithm 1.

---

**Algorithm 1** LLM Data Valuation

---

1: **Input:** Datasets $\{\boldsymbol{S}^{(i)}\}_{i=1}^K$ from $K$ contributors, each consists of real and synthetic mixtures; validation set $\boldsymbol{T}$; model $f$ with initialized NTK kernel matrix $\boldsymbol{\Theta}_0$; weighting coefficients $w_1, w_2, w_3, w_4$.
2: **for** $i = 1$ to $K$ **do**
3:    Evaluate $v(\boldsymbol{S}^{(i)})$ by (8).
4: **end for**
5: **Output:** Valuation scores $\{v(\boldsymbol{S}^{(i)})\}_{i=1}^K$.

---

## C  GENERALIZATION TO MARGINAL EVALUATION

In LLM-scale training datasets, computing marginal contributions through retraining-based methods like leave-one-out or Shapley value becomes computationally costly due to the model size and dataset scale. To address this, we utilize $v(\boldsymbol{S})$ as the data valuation function, following the existing work (Choe et al., 2024; Park et al., 2023).

While this method is our default for LLM-scale applications, our scoring function $v(\boldsymbol{S})$ remains compatible with marginal estimation for general-purpose data valuation scenarios with smaller models or datasets. Given a collection of data contributors $\{\boldsymbol{S}^{(1)}, \ldots, \boldsymbol{S}^{(K)}\}$, we define the marginal contribution of contributor $i$ with respect to a coalition $C \subseteq [K] \setminus \{i\}$ as:

$$\Delta_{i,C} = v(\boldsymbol{S}^C \cup \boldsymbol{S}^{(i)}) - v(\boldsymbol{S}^C), \tag{20}$$

where $\boldsymbol{S}^C = \{\boldsymbol{S}^{(i)}\}_{i \in C}$. The final value of $\boldsymbol{S}^{(i)}$ can then be aggregated over all coalitions via:

$$\phi_i = \sum_{C \subseteq [K] \setminus \{i\}} w_C \times \Delta_{i,C}, \tag{21}$$

where $w_{\mathcal{C}} \geq 0$ denotes coalition weights. In particular, for the SV (Ghorbani & Zou, 2019), $w_{\mathcal{C}} = |\mathcal{C}|!(K - |\mathcal{C}| - 1)!/K!$. For LOO (Koh & Liang, 2017; Koh et al., 2019), $w_{\mathcal{C}} = \mathbb{1}_{\mathcal{C} = \subseteq [K] \setminus \{i\}}$.

## D  EXPERIMENT DETAILS

Here we provide expanded descriptions of the tasks and datasets, baselines, and implementation details that complement Section 5.1.

### D.1  TASKS AND DATASETS

**Image Classification.** We use the CIFAR-100 dataset (Krizhevsky et al., 2009) as the real data, and generate synthetic data by applying corruption transformations from the CIFAR-100-C benchmark (Hendrycks & Dietterich, 2019). These transformations include noise (Gaussian, shot, impulse), blur (defocus, glass, motion, zoom), weather (snow, frost, fog, brightness), and digital (contrast,

elastic, pixelation, JPEG artifacts). We treat each class as a separate data contributor and use retrained accuracy as the ground-truth for evaluation. We construct the dataset using a long-tail distribution over classes, with the frequency of class $i$ is set as $p_i \propto i^{-2}$. Each data contributor is assigned all the data from a single class, resulting in a total of 100 contributors. The proportion of real data for each contributor is fixed at $\pi = 6.25\%$.

**Sentiment Classification.** We use the IMDb (Maas et al., 2011) as the real dataset and the FinGPT Sentiment Train dataset (Yang et al., 2023) as synthetic data. For evaluation, we use the SST-2 (Socher et al., 2013) as the test set. Since the test set contains only positive and negative labels, we filter the training data to include only samples with positive and negative labels, excluding neutral samples, to ensure consistency in the binary classification setup. The evaluation metric is accuracy, calculated as the proportion of test samples where the predicted label matches the ground-truth label. We use 10 data contributors, as detailed in Table 3.

Table 3: Data composition of each contributor $S^{(i)}$ in the sentiment classification task. All sample counts are reported in thousands (k), and each contributor contains 14k samples with varying real data proportion $\pi$.

| Contributor | $S^{(1)}$ | $S^{(2)}$ | $S^{(3)}$ | $S^{(4)}$ | $S^{(5)}$ | $S^{(6)}$ | $S^{(7)}$ | $S^{(8)}$ | $S^{(9)}$ | $S^{(10)}$ |
|---|---|---|---|---|---|---|---|---|---|---|
| Real samples | 9 | 8 | 7 | 6 | 5 | 4 | 3 | 2 | 1 | 0 |
| Synthetic samples | 5 | 6 | 7 | 8 | 9 | 10 | 11 | 12 | 13 | 14 |
| Total samples | 14 | 14 | 14 | 14 | 14 | 14 | 14 | 14 | 14 | 14 |
| $\pi$ | 64% | 57% | 50% | 43% | 36% | 29% | 21% | 14% | 7% | 0% |

**Instruction Following.** We use the Natural-Instructions dataset (Mishra et al., 2021) as the real dataset and the Magpie-Pro-1M dataset (Xu et al., 2024) as the synthetic dataset. The test set is the IFEval benchmark (Zhou et al., 2023). Evaluation is conducted using the IFEval criteria, which include metrics such as instruction-following accuracy—assessing whether the model's output adheres to the instruction's intent, format, and constraints, as defined by a set of predefined rules and templates. We use 4 data contributors, as detailed in Table 4.

Table 4: Data composition of each contributor $S^{(i)}$ in the instruction following task. All sample counts are reported in millions (m), and each contributor contains a different total number of samples with a fixed real data proportion $\pi$.

| Contributor | $S^{(1)}$ | $S^{(2)}$ | $S^{(3)}$ | $S^{(4)}$ |
|---|---|---|---|---|
| Real samples | 0.077 | 0.077 | 0.077 | 0.077 |
| Synthetic samples | 0.180 | 0.077 | 0.033 | 0.009 |
| Total samples | 0.257 | 0.154 | 0.110 | 0.086 |
| $\pi$ | 30% | 50% | 70% | 90% |

**Complex Reasoning.** We use the human-annotated portions of the NuminaMath-CoT training set (Li et al., 2024) as real data and the synthetically generated portions as synthetic data. The test set is the NuminaMath-CoT test set. During training, we perform supervised fine-tuning (SFT) by providing complete reasoning steps and final answers to encourage the model to learn CoT reasoning. For evaluation, we employ a powerful language model *Qwen3-32B* (Team, 2025) as the judgment model to grade the model's output using in-context learning: given the ground-truth reasoning steps and answer alongside the model's output, the judgment model determines correctness. The output is deemed correct only if both the reasoning steps and the final answer match the reference solution. For this task, we construct 100 separate data contributors. Specifically, we partition both the real and synthetic datasets into 5050 equal-sized and non-overlapping partitions. Contributor $i \in \{1, \ldots, 100\}$ is then assigned $(101 - i)$ partitions of real data and $(i - 1)$ partitions of synthetic data, yielding 100 contributors with varying synthetic-data proportions $\pi$.

## D.2 BASELINES

We compare against four representative baselines designed for efficient data valuation. These baselines are selected based on two criteria: (1) they do not require repeated model retraining, making them scalable to LLMs; and (2) they operate with access to checkpoints, gradients, and training/test data.

- **DAVINZ** (Wu et al., 2022), which computes data values from NTK-based approximations at initialization.
- **Deviation** (Lin et al., 2024), which measures deviation in model predictions via kernel ridge regression.
- **LOGRA** (Choe et al., 2024), a label-only gradient attribution method.
- **TracIn** (Pruthi et al., 2020), which tracks training-time gradient similarity.
- **TRAK** (Park et al., 2023), which approximates influence scores using randomized kernel projections.

## D.3 IMPLEMENTATION DETAILS

All image classification experiments are conducted on a single NVIDIA A100 GPU (80GB). All LLM experiments, including sentiment classification, instruction following, and complex reasoning, are conducted on NVIDIA A100 GPUs (80GB each). To combine the four components in our proposed score (Eq.(8)), we treat their respective weights as tunable hyperparameters. In our paper, we optimize the weights $w_1, w_2, w_3, w_4$ by fitting a linear regression, where the target is the average of the empirical loss and the MMD score.

# E SUPPLEMENTARY EXPERIMENTS

This section reports additional experimental analysis complementing the main results, including ranking visualization across data valuation methods.

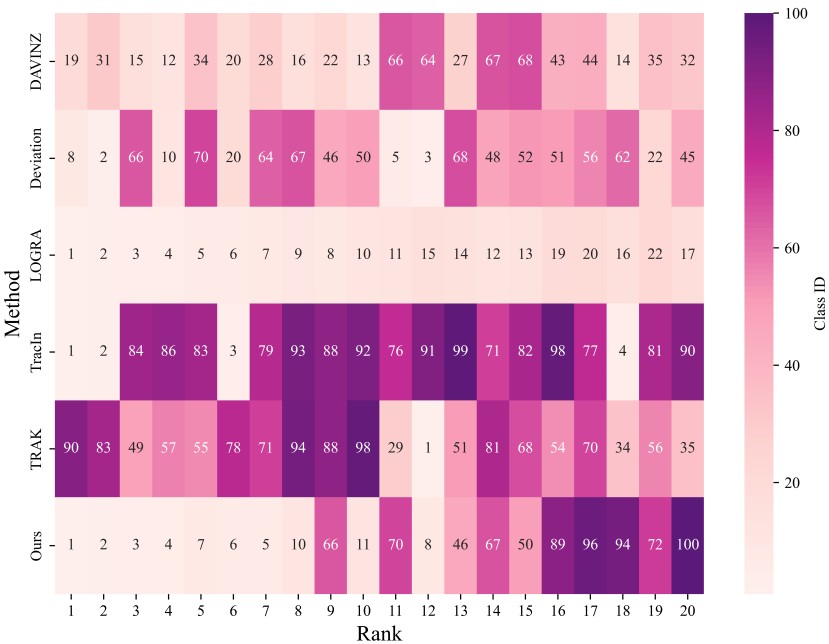

Figure 9: Top-20 contributor selections across data valuation methods on image classification task, where cell colors represent the contributor's class ID.

To complement correlation-based summaries, we visualize the top-ranked contributors selected by each method on the image classification task. In this task, each contributor corresponds to a single

class (100 classes in total), ordered by decreasing sample size (C1 has the most samples while C100 has the fewest). Figure 9 reports the top-20 classes (contributors) deemed most valuable by each data valuation method. We observe that our approach uniquely balances head classes with tail classes, whereas LOGRA, DAVINZ, and Deviation concentrate primarily on head classes, and TracIn and TRAK tend to prioritize tail classes.

## F  THE USE OF LARGE LANGUAGE MODELS

Large language models are employed in a limited and transparent manner during the preparation of this manuscript. Specifically, we use LLMs to assist with grammar checking and minor wording improvements in the writing of this paper. All scientific contributions are entirely the work of the authors.

