# OpenReview forum: "Data Value in the Age of Scaling: Understanding LLM Scaling Dynamics Under Real–Synthetic Data Mixtures"
_ICLR.cc/2026/Conference — ICLR 2026 Conference Withdrawn Submission_

### Official Review · Reviewer_KWsB · 2025-10-19

**Soundness:** 2
**Presentation:** 3
**Contribution:** 3
**Rating:** 4
**Confidence:** 4

**Summary:**

The paper studies how models learn when trained on mixtures of real and synthetic data. It assumes that the underlying knowledge follows a long-tailed distribution, with synthetic data representing a truncated portion of the real distribution. Under this setting, the authors show that training progresses through three distinct phases, corresponding to how the model successively acquires head and tail knowledge.

Building on this perspective, the paper derives a generalisation bound that describes learning behaviour under real–synthetic mixtures. Using the theoretical results, the authors propose a practical, retraining-free data valuation method that estimates the relative importance of different data subsets.

Experiments on long-tailed benchmarks provide partial empirical support for the theoretical analysis and show that the proposed valuation method can identify informative and high-impact data across multiple tasks.

**Strengths:**

* The paper is clearly written and presents a well-structured, well-defined problem formulation.

* The problem formulation is interesting and original, offering a novel way to analyse how real and synthetic data interact during training.

* It tackles a timely and relevant problem, contributing to our understanding of synthetic data in large-scale model development.

* The theoretical analysis is intuitive and well interpreted, giving clear insights into the learning dynamics under long-tailed knowledge distributions.

* The paper includes some empirical evidence on real-world datasets that provides partial support for the theoretical claims and the proposed data-valuation method.

**Weaknesses:**

* The paper does not provide sufficient analysis to justify the central hypothesis that knowledge in real–synthetic mixtures follows a long-tailed distribution and that synthetic data represent a truncated version of it. In real-world settings, the distribution of knowledge can vary substantially across modalities and tasks (e.g. datasets such as CIFAR-100 are relatively balanced). Moreover, model-based synthetic data generation methods often suffer from hallucination or misalignment with real data, and their outputs can be highly sensitive to the training distribution or the prompt used for generation. It would also be valuable to examine, either empirically or theoretically, how sensitive the proposed theoretical results are to deviations from this idealised assumption. For instance, if the synthetic distribution exhibits different levels of noise or if the real data distribution departs from a pure long-tail form. A deeper empirical and theoretical investigation of this assumption would significantly strengthen the paper’s applicability and credibility.

* The analysis of the proposed data-valuation method is limited. The score involves four components with associated weights, yet there is no ablation or sensitivity study showing how the valuation results change when these weights are adjusted or when specific terms are removed. It remains unclear which components are most critical for stable performance.

* The NTK-based term is computed at random initialization, which may exhibit large variance across random seeds or architectures. The paper does not discuss whether this variability could affect the reliability or reproducibility of the valuation results.


* The work focuses mainly on data-attribution–style approaches and overlooks recent data-weighting frameworks that address similar goals. In particular, methods such as [1] and [2] also aim to evaluate or adjust the contribution of data mixtures (at domain or instance level). Interestingly, the first two terms of the proposed score relate to distribution alignment, and the third term addresses redundancy, concepts that also appear in these weighting-based methods. A deeper comparison or discussion would clarify the novelty and positioning of this work.

[1] Xie et al., DoReMi: Optimizing Data Mixtures Speeds Up Language Model Pretraining. NeurIPS 2023

[2] Kuo et al., Not All LLM-Generated Data Are Equal: Rethinking Data Weighting in Text Classification. ICLR 2025

**Questions:**

1. **About Lemma 1 and the loss definition:**
   In Lemma 1, it appears that $\mathcal{L}_{\text{test}}$ should correspond to the *error rate* (as implied by the derivation of lemma1), rather than an arbitrary loss function. However, in both the appendix and the main text, the derivation seems to treat it as a general loss. Could the authors clarify which interpretation is correct, and whether the asymptotic equivalence used in the lemma depends on this choice?

2. **Assumption on $\lambda_{\min}(\Theta_0) > 0$ in Theorem 1:**
   While $\Theta_0$ is symmetric (hence has real eigenvalues), it is only strictly positive definite if the gradient feature vectors of all samples are linearly independent. In practice, however, when some data points are identical or highly similar, these gradients can become correlated, making $\Theta_0$ singular or ill-conditioned. Could the authors discuss how realistic this assumption is for practical networks and whether any regularisation (e.g., adding $\lambda I$) was used to ensure stability or invertibility in their implementation?

3. **Fitting of $w_1, w_2, w_3, w_4$:**
   In the experiments, the weights are learned via linear regression using the average of the empirical loss and the MMD score as the target. Given that this target is directly related to the first two components of the value function, do the fitted results make $w_3$ and $w_4$ nearly zero? Could the authors clarify the rationale for this design choice and whether this regression target effectively serves as a baseline, since it is already part of the objective?

4. **Concept of data value and dependency among contributors:**
   The paper defines data value based on the generalisation bound but does not explicitly account for *dependencies among contributors* or for what knowledge the model has already learned. Is this aspect ignored because the bound is not tight, or is there an implicit assumption of independence between subsets?

In summary, while the paper is clear and conceptually interesting, the justification of key assumptions and the breadth of empirical validation are limited. These issues make the work less convincing as a practical contribution, though it remains theoretically insightful.
If the authors can convincingly address these weaknesses and questions, I believe the paper would be strong enough for acceptance.

---

### Official Review · Reviewer_dKjG · 2025-11-01

**Soundness:** 2
**Presentation:** 2
**Contribution:** 2
**Rating:** 2
**Confidence:** 3

**Summary:**

This work explores how large language models scale when trained on mixtures of real and synthetic data. It proposes a theoretical framework that characterizes a three-phase scaling behavior with two critical breakpoints and derives a generalization bound for such mixtures.  From the bound, the authors propose a data valuation method to estimate the contribution of real and synthetic subsets and validate it by simulations and multiple benchmarks.

**Strengths:**

- The authors try to address a timely and important topic.
- They provide a conceptually novel framework, combining scaling-law intuition with a formal generalization bound and data valuation.
- The efficiency of the proposed approach is supported by experiments.

**Weaknesses:**

Synthetic data is a hot topic, but we should carefully consider what kinds of synthetic data research are genuinely needed and helpful.  The paper raises an interesting perspective but does not clearly articulate what practical insights or actionable implications this line of work contributes to the broader understanding of synthetic–real data mixtures.

- The paper lacks mathematical justification for the two breakpoints in Lemma 1. It is not clear why two critical points are at sample sizes $|S| = k^{\beta}$ and $|S| = k^{\beta}/\pi∣$. There is no analytical explanation or theoretical grounding for why these transition points should arise, nor a deeper discussion of the mechanisms behind the proposed three-phase behavior.

- All visualizations of the “three stages” are based purely on simulation results. No real experimental demonstrations substantiate that this three-phase scaling truly appears in practical model training. Moreover, even within the simulations, the left panel of Figure 6 does not clearly separate the three learning stages.

-  The correlation evaluation between the visualization score and the ground truth needs clearer justification or discussion. The authors need to discuss why test accuracy or other metrics are appropriate proxies for true data importance.

**Questions:**

- In Figure 6, most simulation settings use a very low proportion of real data ($\pi$). Why is the analysis focused primarily on low real-data ratios? What is the benefit or motivation behind this choice? How does it reflect real-world scenarios, and would broader or more balanced real–synthetic ratios lead to different scaling behaviors or insights?
- Do the authors have any selection or filtering process for the synthetic data used in experiments? How do you ensure the quality or representativeness of the synthetic samples, and what measures are taken to prevent low-quality or biased synthetic data from influencing the results?

---

### Official Review · Reviewer_fFvw · 2025-11-01

**Soundness:** 1
**Presentation:** 3
**Contribution:** 2
**Rating:** 4
**Confidence:** 4

**Summary:**

This paper studies how LLM performance scales when training on mixtures of real and synthetic data. It posits a *three‑phase* scaling pattern—Rapid‑Learning, Plateau, and Tail‑Learning—with two breakpoints tied to a truncation of tail knowledge in the synthetic distribution (Figures 2–3). The theory is formalized via **Theorem 1**, a generalization bound that depends on (i) empirical losses on real/synthetic subsets, (ii) distribution discrepancies between train and test, (iii) an NTK‑based term at initialization, and (iv) the real‑data proportion ω and sample size |S| (p. 5). Building on the bound, the authors propose a *retraining‑free* data valuation score (v(S)) (Eq. (8), p. 5) that combines weighted empirical losses, MK‑MMD discrepancies, an NTK term ($ \hat y^\top \Theta_0^{-1} \hat y / |S|$ ), and a small‑sample correction ( \max($\omega,1-\omega)/|S| $). Experiments on CIFAR‑100/100‑C (image classification), IMDb/FinGPT (sentiment), Natural‑Instructions/Magpie (instruction following), and NuminaMath‑CoT (reasoning) report: (a) visual evidence for the three‑phase pattern under a long‑tail setup (Figures 4–6, p. 7), (b) higher correlations to “ground‑truth” contributor value than DAVINZ, Deviation, LOGRA, TracIn, and TRAK (Figure 7, Table 1, p. 8), (c) lower runtime (Figure 8, p. 8), and (d) stability of relative scores under subsampling (Table 2, p. 9).

**Strengths:**

1. **Timely focus and clear problem statement.** The paper tackles a pressing question—how to reason about data value and scaling when synthetic data increasingly dominates LLM training.
2. **Readable presentation of the scaling picture.** The results are demonstrated in a clean and nice ways.

**Weaknesses:**

1. **Experimental Setting is inconsistent with the Theory**
 The core assumption is that the synthetic distribution is *head‑only up to a cutoff k* (p. 4), leading to the two breakpoints. However, the “real vs synthetic” choice in the *image* experiment uses CIFAR‑100 vs CIFAR‑100‑C (corruptions) (p. 6). Corruptions degrade images but do **not** emulate head‑only token sampling or nucleus‑truncation; they don’t reduce class‑tail support in the sense assumed by the theory. This gap makes it hard to view Figures 4–5 (p. 7) as a direct validation of the truncation‑driven breakpoints. A more faithful test would generate synthetic **from a model trained on the real set** (e.g., top‑p/temperature sampling) and control tail truncation explicitly, then mix real and generated data as in the theory.)
2. **Three‑phase evidence needs stronger quantification.**
 The “Plateau” region in Figure 5 (p. 7) is not convincingly flat; visually, the loss keeps trending without a clear slope break. Please quantify both breakpoints (e.g., via change‑point detection or piecewise‑power‑law fits with confidence intervals), report slopes and uncertainty across seeds, and show head/tail decompositions with error bars.
3. **Bound terms need intuition.**
 Theorem 1 introduces two important terms: the initialization‑NTK term ( $\hat y^\top \Theta_0^{-1}\hat y / |S|$ ) and the finite‑sample term ($2\max(\omega,1-\omega)\log(8/\delta)/|S|$) (p. 5). The paper should add intuitive explanations for how each term emerges.
4. **Computational complexity.** Computing ( \Theta_0^{-1} ) is (O(n^3)) naively; at LLM scale this is large.  This may be the reason why text experiments are only performed on LLMs smaller than 7B, which largely limits the practical value of the method.
5. **Assumption setting needs empirical grounding.**
 The model assumes ($p_i \propto i^{-\varepsilon}$) for knowledge frequency. The paper should provide empirical evidence (e.g., token/knowledge histograms) that downstream tasks indeed follow the specified forms or offer more insights on this assumption.
6. **Fairness and statistical power of comparisons.**
 Gradient‑based baselines are restricted to 1% of training data (p. 6), but it’s unclear whether your own method’s NTK/MMD computations were likewise subsampled or ran on full data. Please match compute budgets, report absolute FLOPs/time/memory, and repeat the comparison under equalized budgets.

*I lean toward rejecting this paper, as it still requires refinement. However, I hope the authors do not take this as a harsh judgment—the work has potential, and with further effort, it could evolve into a solid and interesting contribution.*

**Questions:**

See weakness above.

---

### Official Review · Reviewer_iNXG · 2025-11-03

**Soundness:** 3
**Presentation:** 3
**Contribution:** 2
**Rating:** 2
**Confidence:** 5

**Summary:**

The authors are interested in characterizing the generalization behavior of large models trained on mixtures of real and synthetic data. They give two theoretical results.

The first result is based on a stylized model that conceptualizes the real data as consisting of abstract entities called "knowledge" that follow a Zipf distribution. The synthetic data follows a truncated, re-normalized, version of that distribution where tail entities are missing. This is meant to model the fact that synthetic data tends to be better at capturing the head of its distribution. They further assume that the probability the model makes a correct prediction on a knowledge entity is given by a Bernoulli whose parameter takes a specific parametric form, and depends on whether the knowledge was observed in training data, and the rank of the knowledge entity in the real distribution. Given these assumptions, they prove a three-phased scaling law (over the training size) which posits that generalization rapidly decreases, plateaus, before exhibiting further decrease.

The second result states an LLM generalization bound based on Neural Tanget Kernels (Jacot et al), adapted to accommodate a distribution that is a union of two components, the real and synthetic parts.

The authors provide empirical validations for the first setting on CIFAR-100 where the label class is used as the knowledge entity and the distribution is adjusted to fit the modeling assumptions of the setting. They also provide pure simulations that explore the effect of different mixing ratios between real and synthetic data. Finally, they evaluate the second result by predicting the value of different datasets across a varied set of tasks: sentiment classification, instruction-following and reasoning.

**Strengths:**

Strength #1: The paper considers a very interesting a relevant problem. Essentially, how do we adapt scaling laws to account for synthetic data? Compelling results on this subject would be broadly interesting and important to the community.

Strength #2: I appreciate how the paper is written. The ideas are presented clearly and succinctly.

**Weaknesses:**

Weakness #1: The first theoretical result proved by the authors (three phase generalization) relies heavily on a set of very parametric assumptions. While that is not necessarily a fatal flaw, I would have liked more thorough empirical results to validate and support the theoretical finding as a robust scaling law. The existing experiment is a quasi-simulation, where the distributional assumption of the theory is enforced in the sample. It is also conducted on a single task, image classification on CIFAR. Moreover, using the label as the knowledge entity is the setting where I would most expect the presence of knowledge in the data to affect predictive accuracy.

Weakness #2: The proof Theorem 1 is almost identical to that of the proof of Theorem 1 in "DAVINZ: Data Valuation using Deep Neural Networks at Initialization" (Wu, et al). Treating the dataset as a mixture of real and synthetic data does not seem to introduce too much additional technical complexity or require any deep insights. Overall, the contribution here seems very incremental to me.

**Questions:**

The three phase behavior of large-model generalization is frequently observed in practice. An interesting experiment might be to find a dataset where this is observed naturally (without modifying the label distribution to enforce it), and then find a class of knowledge that follows the parametric laws posited by the theory (this should be easy considering the ubiquity of heavy-tailed data phenomena), and finally demonstrate that the transitions observed in practice are consistent with the theory. Have the authors considered extending their work in this direction?

Why is the method so good at the instruction task, often attaining perfect correlation? Is it because there are only 4 contributors?

Minor nits:
Grammar: “an salient data valuation framework”
Grammar: “given the training set S contain”

---

### Note · Authors · 2025-11-19

I have read and agree with the venue's withdrawal policy on behalf of myself and my co-authors.